# Use of Sentinel-1 radar observations to evaluate snowmelt dynamics in alpine regions

Carlo Marin[1], Giacomo Bertoldi[2], Valentina Premier[1], Mattia Callegari[1], Christian Brida[2], Kerstin Hürkamp[3], Jochen Tschiersch[3], Marc Zebisch[1], and Claudia Notarnicola[1]

[1]Institute for Earth Observation, Eurac Research, Viale Druso, 1 I-39100 Bolzano, Italy
[2]Institute for Alpine Environment, Eurac Research, Viale Druso, 1 I-39100 Bolzano, Italy
[3]Helmholtz Zentrum München, German Research Center for Environmental Health, Institute of Radiation Medicine, Ingolstädter Landstraße 1, 85764 Neuherberg, Germany

**Correspondence:** Carlo Marin (carlo.marin@eurac.edu)

**Abstract.** Knowing the timing and the evolution of the snow melting process is very important, since it allows the prediction of: i) the snow melt onset; ii) the snow gliding and wet-snow avalanches; iii) the release of snow contaminants and iv) the runoff onset. The snowmelt can be monitored by jointly measuring snowpack parameters such as the snow water equivalent (SWE) or the amount of free liquid water content (LWC). However, continuous measurements of SWE and LWC are rare and difficult to obtain. On the other hand, active microwave sensors such as the Synthetic Aperture Radar (SAR) mounted on board of satellites, are highly sensitive to LWC of the snowpack and can provide spatially distributed information with a high resolution. Moreover, with the introduction of Sentinel-1, SAR images are regularly acquired every 6 days over several places in the world. In this paper we analyze the correlation between the multi-temporal SAR backscattering and the snowmelt dynamics. We compared Sentinel-1 backscattering with snow properties derived from in situ observations and process-based snow modeling simulations for five alpine test sites in Italy, Germany and Switzerland considering two hydrological years. We found that the multi-temporal SAR measurements allow the identification of the three melting phases that characterize the melting process i.e., moistening, ripening and runoff. In detail, we found that the C-band SAR backscattering decreases as soon as the snow starts containing water, and that the backscattering increases as soon as SWE starts decreasing, which corresponds to the release of meltwater from the snowpack. We discuss the possible reasons of this increase, which are not directly correlated to the SWE decrease, but to the different snow conditions, which change the backscattering mechanisms. Finally, we show a spatially-distributed application of the identification of the runoff onset from SAR images for a mountain catchment, i.e., the Zugspitze catchment in Germany. Results allow to better understand the spatial and temporal evolution of melting dynamics in mountain regions. The presented investigation could have relevant applications for monitoring and predicting the snowmelt progress over large regions.

## 1 Introduction

Seasonal snowpack is one of the most important water resources present in nature. It stores water during the winter and releases it in spring during the melting. In mountain regions, snow storage is essential for the freshwater supply of the lowlands, making

the mountains the water towers of the downstream regions (Viviroli and Weingartner, 2004). In fact, the temporally delayed release of the water from the head-watersheds to the forelands is essential for a large number of human activities such as agriculture irrigation, drinking water supply and hydropower production (Beniston et al., 2018). In particular, in the Alps, discharges in May and June are largely dictated by snowmelt, while from July to September are influenced by glacier melt (Wehren et al., 2010) and liquid precipitation. On the other hand, wet snow may contribute to natural disasters such as wet snow avalanches (Bellaire et al., 2017) or wet-snow gliding (Fromm et al., 2018). Moreover, in case of accumulated contaminant release from a snowpack, initial runoff meltwater can be highly enriched and is able to cause severe impact on the water quality (Hürkamp et al., 2017). In this context, knowing the temporal and spatial evolution of the snow melting process is very important for a proactive management of the water resources and for hazard mitigation.

The melt period can be generally separated in three phases (Dingman, 2015): i) moistening, ii) ripening and iii) runoff. The moistening is the initial phase of the snowmelt. The air temperature and solar radiation increase and due to heat exchanges and/or rain the superficial layers of the snowpack start melting. The ripening phase begins when the maximum retention capacity of the pores is exceeded. The wetting front penetrates through the snowpack, driven by repeated cycles of melting and refreezing, but the meltwater is not yet released. During this phase, the snowpack becomes isothermal and when no more liquid water can be retained, the runoff phase starts. The snowmelt process is a non-linear process affected by the strong variability of both the snowpack characteristics and the meteorological forcings that affect the snow. In order to obtain useful information about the progression of the melting process, non-invasive techniques that allow performing multiple measurements at the same location should be exploited. For this purpose, measurements of meteorological variables such as air temperature, snow temperature, relative humidity, wind speed, precipitation, and solar radiation are usually employed to extract information on snow melt dynamics (Kinar and Pomeroy, 2015). However, the most significant state variables to properly identify the three melting phases are the snow water equivalent (SWE), i.e. the total mass of liquid and solid water stored in form of snow, and the liquid water content (LWC), i.e. the mass of liquid water inside the snowpack. An increase of LWC in time indicates a moistening process going on. The downward penetration of the water front into the snowpack brings first, to a partial, and later to a complete isothermal state. This leads to the generation of water runoff, and cosequently to a significant decrease of SWE.

Continuous measurements of SWE and LWC is therefore essential to monitor the snowpack melting dynamics. So far, the most common method to manually measure SWE is using snow sampling tubes, while the most spread techniques for automatic SWE measurement include snow pillows and snow scales (Kinar and Pomeroy, 2015). The installation and the maintenance of these kinds of measurements are very costly and a relatively limited number of continuous measurements of SWE are available in the Alps. Direct measurements of LWC are usually performed through empirical estimations (e.g. the hand test), or indirect assessments based on snow temperature. Recently, some promising systems that exploit the dielectric properties of the snow in the microwave region of the electromagnetic (EM) spectrum have been presented to allow a continuous and nondestructive measuring of LWC. In particular, three systems have demonstrated to be effective and robust in operational conditions: i) the snowpack analyzer (SPA) (Stähli et al., 2004); ii) the snow sense (Koch et al., 2014) based on GPS signals; and iii) the upward-looking Ground Penetrating Radar (upGPR) (Schmid et al., 2014). All of them are commercial systems buried under the snowpack and rely on different methods for the dielectric constant estimation. Interestingly, these EM devices

can be used to measure the SWE as well. However, all these ground-based measurements are limited in application to a single point, require calibration to relate the dielectric constant to volumetric snow LWC, and some of them are expensive, power intensive and laborious to be installed and maintained. These limitations complicate the possibility to monitor and understand the meltwater runoff and the snow stability considering also the spatial variability of the snowmelt dynamics.

To mitigate these limitations, energy based, multilayer physically based snow models can simulate SWE and LWC at high spatial and temporal resolution (Essery et al., 2013). Such kind of models account for shading, shortwave and longwave radiation, and turbulent fluxes of sensible and latent heat (Mott et al., 2011), but can differ in the way they parametrize snow metamorphism, grain size evolution, snow layering and liquid water percolation (Wever et al., 2014). They can range from very detailed approaches with a Lagrangian representation of snow layers as avalanche-forecasting models like CROCUS (Brun et al., 1992) or SNOWPACK/ALPINE3D (Bartelt and Lehning, 2002; Lehning et al., 2006) to more simplified approaches as the ones of hydrologically-oriented Eulerian models as AMUNDSEN (Strasser et al., 2011) or GEOtop (Endrizzi et al., 2014). Therefore, snow models can provide detailed information about the snow properties starting from observed meteorological conditions, which can be reliably acquired especially at plot-scale. However, model performances are affected by uncertainties and errors related to model structure (Avanzi et al., 2016), meteorological forcing (Raleigh et al., 2015) and model parametrizations (Engel et al., 2017; Günther et al., 2019). Therefore, there is the need of snow observations with high temporal and spatial resolution, distributed over a large area and systematically acquired.

In the past years, Synthetic Aperture Radar (SAR) was shown to be a valid tool to identify the wet snow i.e., snow that contains a given amount of free liquid water (Nagler and Rott, 2000; Dong, 2018). In fact, SAR measurements are highly sensitive to the liquid water in the snowpack and the increase of the LWC causes a high dielectric loss that increases the absorption coefficient generating backscattered signal with low intensity (Ulaby et al., 2015). This physical principle has been exploited for the generation of wet snow maps by the bi-temporal algorithm proposed by (Nagler and Rott, 2000) and further improved in (Nagler et al., 2016). However, the increase of the liquid water content explains only partially the decrease of the backscattering coefficient. Indeed, as pointed out in Shi and Dozier (1995) and Baghdadi et al. (2000), the relationship between the coefficient of backscattering and the snow wetness can cause an increment of the backscattering value depending on the conditions of the snow roughness, snow density, snow layering, snow grain size and local incidence angle. This large number of unknowns, upon which the SAR backscattering is dependent on, defines a complex multiparametric problem that is difficult or even impossible to solve without introducing some simplification assumptions. So, even though some works have been presented that try to extract the LWC using C-band SAR images (Shi and Dozier, 1995; Longepe et al., 2009), at the best of our knowledge there are no attempts to use the SAR as source of information for describing the multi-temporal evolution of the snow melting process. Progress has been hampered by: i) the lack of ground truth information; ii) the relative high number of sources of uncertainty of the SAR signal; and iii) the difficulty to access SAR data in the past. This has changed since 2014 with the introduction of the Sentinel-1 (S-1) mission from the European Space Agency (ESA) and the European Commission (EC) guarantying the availability of C-band SAR images free of charge. In detail, S-1 is a constellation made up of two near-polar sun-synchronous satellites that acquire images early in the morning and late in the afternoon, with a revisit time of 6 days at the equator. Moreover, as discussed before, an increasing number of data on relevant snow parameters related to the

snowmelt are collected by operational systems (e.g. by SPA) or derived by physically based snow models. The information on SWE and LWC provided by independent sources opens new opportunities for better understanding the relationship between the snowpack properties during the melting phase and the multi-temporal SAR backscattering.

The aim of this work is to evaluate the information that S-1 can provide on monitoring the snowmelt dynamics. In particular, we provide the theoretical EM background for understanding the impact on the multi-temporal SAR backscattering of a melting snowpack. Then, we analyze the relationship between the multi-temporal SAR signal acquired from S-1 and in situ measurements of LWC and SWE in the Alps. Given the limited number of point-related continuous SWE and LWC measurements available in the test area, we made use of the physically based model SNOWPACK to simulate the snow properties in other locations where only meteorological data and snow depth were available. This allowed us to define five test sites at different altitudes in the Alps, where the interactions of S-1 backscattering with the snowpack were studied in detail during two melting seasons. On the basis of the outcomes of the study, we propose an interpretation scheme to be applied to multi-temporal dual polarization C-band SAR data in order to identify the different snow melting phases of moistening, ripening and runoff. Finally, we demonstrate the effectiveness of the proposed approach in a real application scenario to provide a spatially distributed information about the melting phases of the snowpack in alpine terrain, which can be used for monitoring and predicting the snowmelt progress over large regions.

## 2 Background

In this section we report the theoretical background on which this work is based on. First, the snow melting process is explained from a physical point of view and the different phases are identified considering the information of LWC and SWE. Then, the response of the SAR backscattering to the wet snow is described in detail.

### 2.1 Snow melting process

Figure 1 illustrates the snow cover development during the melting season considering the snow status in the morning and in the afternoon, when the S-1 descending and ascending data is acquired respectively. Hypothetical values of LWC and SWE are reported on the right side of the figure. In general, the liquid water is introduced in the snow by rain and/or melt due to heat exchange and the incoming flux of shortwave radiation flux, which varies with slope, aspect and elevation. In both cases, the snowpack starts melting at the surface (Techel and Pielmeier, 2011). This superficial moistening phase can be identified by comparing observations from the coldest and warmest period of the day i.e., a diurnal cycle is visible. Interestingly, the SAR acquisitions are approximately acquired around these two periods. The liquid water released or absorbed from the superficial layers gets in contact with the subfreezing snow present underneath and freezes. This releases latent heat that causes the snowpack to warm up starting the process of snow ripening. Repeated cycles of partial melting during the day and refreezing during the night induce the development of the wetting front into the snow. This is generally not uniform, since infiltrations usually start through isolated "flow fingers" which enlarge into meltwater channels due to the passing of time. Therefore, the ripening of the snowpack may be different year by year or considering different areas. In fact, climatic factors or snowpack

stratifications may induce different behaviors. At the point of full water saturation, the snow layer cannot retain any more liquid water. Further absorption of energy produces water output, which, depending on soil properties, ice and water content, could infiltrate in the soil or appear as surface runoff (DeWalle and Rango, 2008). The runoff phase is characterized by a significant decrease of SWE.

     During the melting, the presence of liquid water inside the snowpack directly affects the grain size, the grain shape and the
density of the pack (Pomeroy and Brun, 2001). Indeed, during the melt process the snow undergoes to a rapid metamorphism that leads to a growing and a rounding of the grains linked to an increase of the snow density. Moreover, it is important to underline that during the melt season a general increase of the roughness of the snow surface is observed (Fassnacht et al., 2009) due to localized melting pattern (i.e., flow fingers) and rain on snow events.

## 2.2    SAR backscattering response to wet snow

From an EM point of view, the snowpack is an inhomogeneous medium composed of scattering elements with different sizes, shapes, orientations and permittivity values. The backscattering $\sigma^0$ produced by an EM wave generated by SAR over such a medium can be modeled as an incoherent sum of three contributions (Shi and Dozier, 1995; Ulaby et al., 2015): the surface scattering produced at the air-snow interface, $\sigma_{sup}^0$, the surface scattering produced at the snow-ground interface attenuated by the snowpack, $\sigma_{grd}^0$, and the volumetric scattering of the snowpack, $\sigma_{vol}^0$. The intensity of these contributions depends on
parameters related to: i) the sensors i.e., frequency, local incidence angle (LIA) and polarization; ii) the snowpack properties i.e., liquid water content (LWC), density (DS), ice particle size and shape (GS), surface roughness (RS), which is usually described by the standard deviation of the height and the correlation length of the surface; and iii) the ground properties. In this paper we focus on the use of the C-band SAR mounted on board of S-1, and therefore all the parameters related to the sensor are known. Nonetheless, deriving the theoretical behavior of the time series of $\sigma^0$ for a given LIA for one hydrological
145    year is complex. Indeed, the relationship between the backscattering and the snow parameters forms a non-linear system of equations. In the following we identify the main scattering mechanisms isolating the contribution of each parameter to the total backscattering.

     During the accumulation period, dry snow is almost transparent for C-band, and the radar echo can penetrate the snow for several meters. In this situation, the main scattering source is the snow-ground interface (see Fig. 2) and the backscattering is
almost insensitive to different snow parameters (Rott and Mätzler, 1987; Shi and Dozier, 1993). During the melting period, the increase of the free liquid water inside the snowpack causes high dielectric losses, which increase the absorption coefficient. By considering a sufficiently thick snowpack, this leads to a rapid decrease of $\sigma_{grd}^0$, which can be then neglected. By assuming constant all the parameters but the LWC, the increase of LWC causes the volume scattering to decrease and the backscattering becomes sensitive to surface roughness (Shi and Dozier, 1995). When the surface is smooth e.g., according to the Frauenhofer
criterion (Ulaby et al., 2015), volume scattering dominates and therefore the increase of LWC results in a decrease of the total backscattering. Whereas, when the surface is rough the surface scattering dominates, thus with the increase of LWC the total backscattering tends to increase. The amount of wetness from which the surface scattering becomes predominant depends mainly on the surface roughness and LIA and may vary from about 1% to 6% of the total volume (Magagi and Bernier, 2003).

However, other parameters play a role in this mechanism: by assuming constant all the parameters but the snow density, the volume scattering decreases at the increase of the snow density, if all the other parameters are kept fixed. Vice versa, the grain size increases the volume scattering. It is finally worth stressing the fact that the response to the wet snow becomes more complex in case of the snowpack in forest (Koskinen et al., 2010). In this case the total backscattering $\sigma^0$ is a function also of the forest stem volume. This can be estimated and taken into account, nonetheless in this work we focus on the identification of snow melting phase in open areas.

The main scattering mechanisms and their influence on the backscattering, as studied in the literature, are reported in Table 1. Even though the table is reporting the main backscattering mechanisms of the different snow conditions during the melting process, the complete multi-temporal behavior that characterizes the three phases of moistening, ripening and runoff has not yet been studied. In particular, from an EM modeling point of view or real-data analysis, the implications of the wet-snow metamorphism i.e., increase of LWC, density, snow grain size and superficial roughness remain mainly unsolved. Indeed, state-of-the-art radiative transfer (RT) models, particularly designed for studying the snow melting process, such as Shi and Dozier (1995); Nagler and Rott (2000); Magagi and Bernier (2003), are not able to model the microstructure scattering interactions. Whereas, RT models that take into account the microstructure interactions, such as for example the modeles developed in SMRT (Picard et al., 2018) or MEMLS3&a (Proksch et al., 2015) are not able to model the contribution from the superficial roughness and have never been specifically tested for the characterization of the melting phases. Therefore, without further research and validation activities, this invalidate the possibilty to use state-of-the-art RT models to better understand the multi-temporal EM mechanisms during the snowmelt at C-band (e.g., Veyssière et al. (2018) found a significant deviation between observations and simulations with MEMLS3&a during the melting period).

In the following, as first attempt to fill this gap, we will consider the real time series of backscattering recorded by S-1 during two hydrological years in the proximity of five test sites where LWC and SWE were measured or simulated. The outcome of this study will be exploited to: i) understand if a characteristic relation can be recognized from the comparison between the multi-temporal SAR signal and the melting phases; and ii) define some rules to automatically identify the beginning of each melting phase from the time series of $\sigma^0$.

## 3    Dataset description

In this section, we present the experimental sites and we describe the collected in situ data, the SNOWPACK set up and S-1 data.

### 3.1    Test sites description, and in situ data

For ground truth and as input for the simulations with SNOWPACK, we consider five snow and meteorological weather stations with different location in terms of place and altitude in the European Alps, equipped with different installed sensors. Among these, one is located in Bavaria (Germany), three in South Tyrol (Italy) and one in Graubünden (Switzerland). In detail, considered parameters are wind velocity (VW), wind direction (DW), air temperature (TA), relative humidity (RH), snow depth (HS),

snow temperature at different depths (TS), surface temperature (TSS), soil temperature (TSG), incoming shortwave radiation (ISWR), incoming longwave radiation (ILWR), outgoing shortwave radiation (OSWR), snow water equivalent (SWE), snow density (DS), liquid water content (LWC) and ice content (IC). The considered data records started from the October 1, 2016 in order to cover the two winter seasons 2016/2017 and 2017/2018. An overview of the location of the stations is presented in Figure 3 and a summary with the available parameters is presented in Table 2.

### 3.1.1  Zugspitze (Werdenfelser Alps, Germany)

The station is located in the Northern Calcareous Werdenfelser Alps, being part of the Zugspitze massif. It is part of the snow monitoring stations network of the Bavarian Avalanche Warning Service (Lawinenwarnzentrale Bayern) and located on a flat plateau at the southern slope of Mt. Zugspitze summit (2962 m a.s.l.), the so-called Zugspitzplatt (1500-2700 m a.s.l.), which is surrounded by several summits in the north, south and west and drained by the Partnach River to the east. Beside a standard meteorological station, the site is additionally equipped with a snow scale and a snow pack analyzer (SPA) to record SWE, DS, LWC and IC. The SPA uses a time-domain reflectometry (TDR) at high frequencies and a low-frequency impedance analyzer. By exploiting different frequencies, the SPA is able to determine the volumetric ice, air and water content as well as the density by measurement of the complex impedance of the snow layer. The EM pulse propagates along three 5 m long sensor bands, horizontally installed in 10 cm, 30 cm and 50 cm above ground in 2016/2017. In 2017/2018 the heights of the bands were changed to 10 cm, 20 cm and 30 cm due to a frequent failure of the uppermost sensor in the preceding years. This allows the measurement of the bulk properties of the snowpack rather than a point measurement as well as a tracking of the downward penetrating water front inside the snowpack. Combined with information on the snow height bulk, LWC is determined. The SPA has not been calibrated for the test site, but it is used with standard set-up parameters and an internal calibration by the manufacturer. This results in unreliable LWC values of about 2-3 % when the snowpack is dry. Moreover, given that no bulk information of LWC for the total thickness of the snowpack is provided by the SPA, we did not use the SPA LWC in this study. Snow height is recorded by an ultrasound sensor, installed at 6 m height. The sensors for the meteorological parameters are installed at a crossbar of the 6 m mast, too, besides the wind sensor, which is at 6.5 m height. The maximum snow height was 3.3 m during winter 2016/2017 and 3.9 m in January 2018. The area is continuously covered by snow between December and May each year. During the accumulation period, the stations records showed that no significant snowmelt runoff at the snow base occurred at any time since 2012 (Hürkamp et al., 2019). During the observed winter seasons the mean monthly wind velocity exceeded $3 \mathrm{~ms}^{-1}$ in the winter months, therefore wind drift could likely alter snow accumulation. The amount of mean annual precipitation is  2000 mm.

### 3.1.2  Alpe del Tumulo (South Tyrol, Italy)

The station is located on an alpine pasture in the North of Val Passiria. For this and the other South Tyrolean stations, the temperature sensor is installed at a 2.8 m height and the wind sensor at 5.5 m. The site is weakly windy, with mean monthly velocity usually around $2 \mathrm{~ms}^{-1}$. The maximum snow height was around 1.5 m during winter 2016/2017 and around 2 m during

the winter 2017/2018. No continuous measurements of LWC and SWE are available for this and the other South Tyrolean stations.

### 3.1.3 Clozner Loch (South Tyrol, Italy)

The station is located in Lauregno (Alta Val di Non) on an almost flat site. The mean monthly wind velocity seldom exceeds 2 $\mathrm{ms}^{-1}$. The snow height never exceeded 1 m during the winter 2016/2017 and the maximum height reached during the winter 2017/2018 was around 1.5 m.

### 3.1.4 Malga Fadner (South Tyrol, Italy)

The station is located on an alpine pasture in Valle Aurina. The mean monthly wind velocity never exceeds 2 $\mathrm{ms}^{-1}$. The maximum snow height was less than 1.5 m during winter 2016/2017 and around 2 m during the winter 2017/2018.

### 3.1.5 Weissfluhjoch (Graubünden, Switzerland)

The automatic weather station is located at Weissfluhjoch, Davos, Switzerland. It is mantained by the WSL Institute for Snow and Avalanche Research SLF. The data are regulary updated and made freely available (WSL Institute for Snow and Avalanche Research SLF, 2015). The wind sensor is intalled at 5.5 m and the temperature sensor at 4.5 m. The site is quite windy, with mean monthly velocity usually around 2 $\mathrm{ms}^{-1}$ or sometimes greater than this value. The maximum snow height was around 2 m during winter 2016/2017 and around 3 m during the winter 2017/2018. In this study, SWE GPS-derived measurements are used (Koch et al., 2019), which are also freely made available upon request.

## 3.2 SNOWPACK model set up

As described in the introduction, the proper identification of the melting phases requires a precise knowledge of the evolution of LWC and SWE. However, these parameters are not always available for the selected test sites. For this reason, there is the need to set up snowpack simulations for obtaining the missing parameters. In this work we used the physically-based model SNOWPACK, a one-dimensional (1-D) model developed by the WSL Institute for Snow and Avalanche Research, SLF (Bartelt and Lehning, 2002). The model solves 1-D partial differential equations governing the mass, energy and momentum conservation. Heat transfer, water transport, vapor diffusion and mechanical deformation of a phase changing snowpack are modeled assuming snow as a three-component (ice, water and air) porous material. Meteorological data are used as input for the model. Required parameters are air temperature, relative humidity, wind velocity, incoming longwave radiation and/or outgoing shortwave radiation, incoming longwave radiation and/or surface temperature, precipitation and/or snow depth and soil temperature. The data were taken or derived from the in situ measurements at the test sites. Meteo-IO (Bavay and Egger, 2014) is used as pre-processing tool to check erroneous data, fill the gaps and generate missing parameters. In the current case, the ground temperature is generated as a constant value assumed to be equal to the melting temperature if missing, and the incoming longwave radiation is calculated through an all-sky parametrization, which makes use of air temperature and humidity

(Unsworth and Monteith, 1975; Dilley and O'brien, 1998). Fresh snowfall must be provided as initial condition. Since direct snow precipitation measurements are not available, the amount of new snow is forced by subtracting the model snow depth to the measured snow depth. This difference is assumed to be fresh snow only if reliable humidity and temperature conditions are verified, using the approach proposed and validated by (Mair et al., 2016) and implemented in the SNWOPACK model. This approach has been validated against snow pillow observations and resulted more reliable compared to heated tipping bucked rain gauges, which may underestimate solid precipitation up to 40% (Sevruk et al., 2009). The energy exchanges on the snowpack surface are imposed either using a Neumann boundary condition (BC), i.e. the energy fluxes are forced, or a Dirichlet BC, i.e. imposing the surface temperature except during ablation when again a Neumann BC is imposed. Additionally, a Dirichlet BC is imposed at the ground interface. A neutral atmospheric surface layer using the Monin – Obukhov similarity theory is imposed. The used water transport model is the NIED scheme proposed by Hirashima et al. (2010). A typical time step of 15 minutes is used for the simulations.

Since the SNOWPACK simulations are used in this work as reference data to be compared against the SAR backscattering, we calibrated the model considering the best agreement in the analyzed years 2016-2018 with in situ snow depth, snow temperatures at three different depth TS1 (0 m from the ground), TS2 (0.2 m from the ground) and TS3 (0.5 m from the ground) and SWE, when available. Pearson correlation coefficient ($\rho$) and the mean absolute error (MAE) have been computed for these variables. Roughness is used as calibration parameter. The results are reported in Table 3.

### 3.3 Remote sensing observations

S-1 is a two satellites constellation with a revisit time of 6 days with the same acquisition geometry and able to acquire dual polarimetric C-band (central frequency of 5.405 GHz) SAR images with a nominal resolution of $2.7 \times 22$ m to $3.5 \times 22$m in Interferometric Wide swath mode (IW). S-1 works in a pre-programmed way in order to build a consistent long-term data archive of images all around the world. IW acquisitions have a swath of about 250 km. This, together with the cycle length of the satellites of 175 orbits, allows the acquisition of more tracks over a given location at the middle latitudes such as the Alps. Therefore, in 6 days more than one acquisition may be available for the area of interest. Table 4 indicates the most relevant parameters related to the data acquisition for each of the selected locations. For the five test sites a total of about 1300 acquisitions were considered. The data used for the presented study are Level-1 ground range detected data, consisting of focused SAR data that have been detected, multi-looked and projected to ground range using an earth ellipsoid model by the data provider. The resulting products have approximately square spatial spacing of 10 by 10 meters. Phase information is lost for this data. This data can be downloaded free of charge form the Copernicus data hub (https://scihub.copernicus.eu/). In order to correct the complex topographic terrain, typical of mountain regions, and to reduce the speckle noise that affects SAR acquisitions, a tailored pre-processing has been applied for all the analyzed data. In detail, the pre-processing operations are performed using the tools included in SNAP (Sentinel Application Platform) version 6.0 and some custom tools developed in Python. In detail, the S-1 backscatter pre-processing operations are the following (S indicates SNAP tool, C indicates custom tool): 1) application of the precise Sentinel orbit to the data (S); 2) removal of the thermal noise present in the images (S); 3) removal of the noise present at the border of the images (C); 4) beta nought calibration (S); 5) assembly of the S-1-tiles coming

from the same track (S); 6) co-registration of the multi-temporal images (S); 7) multi-temporal filtering with a window size 11x11 pixels (C); 8) gamma-MAP spatial filtering 3x3 pixels (S); 9) geo-coding and sigma nought calibration (S); 10) masking of the layover and shadow by considering the local incidence angle (LIA) for each pixel (C). It is worth noting that we use the multi-temporal filter proposed by (Quegan and Yu, 2001). This filter, which is suited for long time-series, allows a suppression of the speckle noise by preserving at the same time the geometrical detail. The final spatial resolution of the geo-coded S-1 images is 20 by 20 meters.

## 4  Data analysis and proposed approach to the melting phases identification from S-1

In this section, the time-series of SWE, LWC and $\sigma^0$ for the identification of the melting phases are compared. From this analysis and the background information described in section 2, we present the general temporal evolution of the backscattering during the melting process. Finally, on the basis of this analysis we propose a set of simple rules for the derivation of the onsets of each snow melting phase.

### 4.1  Data analysis

Figure 4 shows the time series of the backscattering coefficient against the measured and/or modeled SWE and LWC for the five test sites during the hydrological years 2016/2017 (left column) and 2017/2018 (right column). Yellow, red and green areas highlight the moistening, ripening and runoff phases respectively. These phases have been identified from the SWE and LWC data according to section 2.1. In detail, the moistening phase onset is identified by looking at the liquid water content (LWC) of the snowpack. We empirically established a threshold of $1 \mathrm{~kg/m^2}$ that has to be satisfied for at least two consecutive days. In other words, a significant melting (and refreezing) cycle should be observed within two days. Among all the isolated moistening events, in this work we focus only on the moistening preceding a ripening phase. However, this does not mean that the SAR cannot detect isolated peaks of melting, if the acquisitions are performed simultaneously to those events. Regarding the ripening phase, we impose the rule to observe an increase of LWC exceeding $5 \mathrm{~kg/m^2}$ and not decreasing to $0 \mathrm{~kg/m^2}$ during the diurnal cycles. If the LWC returns to $0 \mathrm{~kg/m^2}$ for a timing of at least 5 days, we assume that the ripening phase is interrupted. Otherwise, we assume that there is enough penetration of the waterfront into the snowpack to initiate the ripening. Finally, the runoff phase is identified when SWE starts decreasing from its maximum (after the ripening phase is activated). In the case we have both measured and modelled SWE available, we consider measured SWE as reference. The runoff phase

ends when SWE has a value of $0 \text{ kg/m}^2$. The rules are shown in pseudocode in Algorithm 1.

---

**Algorithm 1:** Identification of the melting phases

**Input:** Liquid Water Content $LWC$ and Snow Water Equivalent $SWE$ observations for a given day

$d$, $d \in \{1, 2, ..., D\}$ with $D$ total number of days with $SWE > 0$, $SWE_{max}$

**Output:** Onset moistening $T_M$, onset ripening $T_R$, onset runoff $T_{RO}$

**while** $d \leq D$ **do**

    **if** $LWC_{max,d} > 0 \ kg/m^2$ **then**

        \# Snowpack is wet

        \# **Check moistening phase**

        **if** $(LWC_{max,d} > 1 \ kg/m^2)$ **and** $(LWC_{min,d} = 0 \ kg/m^2)$ *for at least 2 days* **then**

            $T_M = d$

            \# **Do not check this condition anymore**

            **continue**

        **end**

        \# **Check ripening phase**

        **if** $(LWC_{max,d} > 5 \ kg/m^2)$ **and** $(LWC_{min,d} > 0 \ kg/m^2)$ **then**

            $T_R = d$

            \# **Do not check this condition anymore**

            **continue**

        **end**

        \# **Check runoff phase**

        **if** $(SWE_d == SWE_{max})$ **then**

            $T_{RO} = d$

            \# **Do not check this condition anymore**

            **continue**

        **end**

    **end**

    **else**

        \# Snowpack is dry

    **end**

    d ++

**end**

---

In the following, for each of the five test sites i.e., Zugspitze, Alpe Tumolo, Clozner Loch, Malga Fadner and Weissfluhjoch, we will present the detailed comparison of LWC, SWE and the S-1 $\sigma^0$ time series during the melting process. This will allow the derivation of important information about the possibility to identify the three melting phases in general. In the next section,

the outcome of this comparison will be exploited to describe the characteristic behavior of the multi-temporal SAR signal during the melting process.

### 4.1.1 Zugspitze

For this station, SWE was both measured and simulated and LWC was simulated with SNOWPACK. The temporal evolution of SWE measured by the snow scale and the one simulated with SNOWPACK shows a good agreement. For this station, the tracks T168 (descending, morning) and T117 (ascending, afternoon) are available. The local incidence angle for the two tracks differs of about 1 degree. For the hydrological year 2016/2017 the backscattering remains almost constant during the accumulation phase till the beginning of the moistening phase (Figure 4a). Here, as described in section 2.2, the increase of the LWC is accompanied by a decrease of the backscattering from -8.5 dB and -12.7 dB to -14.3 dB and -20.0 dB for respectively VV and VH of the afternoon track T117 between the 19th and the 25th of March 2017 and from -5.8 dB and -12.7 dB to -12.5 dB and -18.1 dB for respectively VV and VH of the morning track T168 between the 27th of March and the 4th of April. The difference in the dropping of the signal acquired by the morning and afternoon track is due to the diurnal melting and refreezing cycles. After this phase, the ripening phase began with oscillations of the backscattering coefficient which on average presented low values. As described in section 2.2 the oscillations are due to the snowpack metamorphism, snow stratification and the meteorological conditions. Since the ripening phase is characterized by an increase of the LWC, the time series of the backscattering presents a decreasing trend. Interestingly, the minimum of $\sigma^0$ is reached in correspondence to the finishing of the ripening phase and the beginning of the runoff phase i.e., 20th of May 2017. The runoff is instead characterized by a monotonic increase of the backscattering till all snow is melted. This characteristic behavior can be interpreted as follow: when the considered snowpack reaches its saturation condition in terms of the LWC, snow density and internal structure, the backscattering recorded in C-band reaches its minimum value. These snowpack conditions seems to represent the isothermal condition before the release of melt water i.e., the end of the ripening phase. After the saturation point is reached, the monotonic increase of $\sigma^0$ could be explained by a dominance of the superficial scattering that becomes more and more prominent due to a monotonic increase of the LWC per volume (see section 2.2). This behavior continues until the snow disappears. This period corresponds to the runoff formation phase, when SWE starts decreasing. In section 4.2 we will discuss a possible explanation of this apparently surprising behavior. Regarding the winter 2017/2018 similar observations were made, but here the snow ripening phase was limited to a very short period and the runoff started very early in mid-April due to strong insolation and high mean daily temperatures up to 5°C the days before. Interestingly, during the runoff phase, $\sigma^0$ started increasing as expected, then it decreased in correspondence of a snow fall (probably wet) followed by a relatively colder period which lasted some days at the end of May 2018 and finally it increased again till the end of the snow season (Figure 4b).

It is worth noting that the two polarizations acquired by S-1 provided coherent information. However, few cases in which there is a depolarization of the signal can be spotted during the ripening phase. Here the repeated cycles of melting and refreezing can generate ice layers (Kattelmann and Dozier, 1999), which affect the polarization in different ways.

### 4.1.2 Alpe del Tumulo

For this station, the information about the LWC and SWE were derived through SNOWPACK. The calibration of the model was performed in order to achieve a high agreement in terms of snow height and snow temperature (see Table 3). For this station, the tracks T168 (descending, morning), T117 (ascending, afternoon) and T095 (descending, morning) are available. The LIAs for the three tracks are 40, 35 and 47 degrees, respectively.

A very short moistening phase can be identified in both years from the modeled LWC and SWE time series (Figure 4c, 4d). These phases are well identified in the $\sigma^0$ time series by a drop of the morning and afternoon signal. The situation of the runoff phase 2016/2017 looks similar to Zugspitze for the season 2017/2018: from the LWC and SWE time series two modes are visible suggesting that the runoff was stopped by a cold period (with a new snowfall). This situation is reflected in the time series of the S-1 backscattering by the two characteristic "U-shaped" behaviors indicating that a first runoff started after the first minimum of $\sigma^0$ and continued for some days in correspondence of the monotonic increase of $\sigma^0$, but then the process was stopped by a new wet snowfall that forced the backscattering to a new minimum. Finally, the runoff phase restarted, and the SAR signal increased again. However, the runoff phases identified from the SAR local minima seem to be anticipated by about two weeks with respect to the modeling results. Regarding the season 2017/2018, the runoff phase showed a more linear behavior which is represented by the characteristic shape of $\sigma^0$ time series as the one identified in the Zugspitze test site. It is finally worth noting that, the three tracks (T095 and T168, descending, and T117, ascending) acquired with different LIA show very similar trends.

### 4.1.3 Clozner Loch

For this station, the information about the LWC and SWE were simulated with the SNOWPACK model. The calibration of the model was performed in order to achieve a high agreement in terms of snow height and snow temperature (see Table 3). The tracks T168 (descending, morning), T117 (ascending, afternoon) and T095 (descending, morning) are available for this station. The LIAs for the three tracks are 43, 36 and 39 degrees, respectively.

The season 2016/2017 is characterized by two melting phases (Figure 4e). In fact, the snow was completely melted in the first half of April with a new fresh snowfall at the end of the month. For this reason, we highlighted two different times the snowpack alteration sequence moistening – ripening – runoff. Interestingly, the time series of the backscattering seems to properly follow the two melting processes with two characteristic "U-shaped" behaviors. The melting process for the season 2017/2018 was more linear (Figure 4f) and the $\sigma^0$ time series of the three tracks provides coherent information with the one extracted by analyzing the time series of LWC and SWE.

### 4.1.4 Malga Fadner

For this station, the information about the LWC and SWE were derived through the SNOWPACK model. The calibration of the model was performed in order to achieve a high agreement in terms of snow height and snow temperature (see Table 3). Four

tracks are available for this station: T168 (descending, morning), T117 (ascending, afternoon), T044 (ascending, afternoon) and T095 (descending, morning). The LIAs for the three tracks are 46, 48, 38 and 34 degrees, respectively.

The trend of the melting process over the two seasons looks similar to Alpe del Tumulo. The season 2016/2017 is characterized by a consistent snowfall, which happened after an initial runoff phase of the snowpack. This together with a cold period,
stopped the process, which was resumed in May (Figure 4g). The time series of the four tracks recorded by S-1 backscattering showed two characteristic "U-shaped" behavior indicating that a first runoff started after the first minimum of $\sigma^0$ and continued for some days in correspondence of the monotonic increase of $\sigma^0$, but then the process was stopped by a new wet snowfall that forced the backscattering again to the minimum. Nonetheless, the timings are different from the one identified with the modeled data of LWC and SWE. The strong depolarization may indicate a complex structure of the snowpack with different
ice layers. The melting process for the season 2017/2018 was more linear and the $\sigma^0$ time series of the four tracks provides coherent information with the one extracted by analyzing the time series of LWC and SWE (Figure 4e).

### 4.1.5 Weissfluhjoch

For this station, the information about the LWC and SWE were simulated with SNOWPACK, additionally SWE GPS-derived measurements were available. The calibration of the model was performed in order to achieve a high agreement in terms of
snow height and SWE (see Table 3). The tracks T168 (descending, morning), T117 (ascending, afternoon), T015 (ascending, afternoon) and T066 (descending, morning) are available for this station. The LIAs for the three tracks are 41, 33, 43 and 31 degrees, respectively.

The season 2016/2017 is characterized by an initial moistening phase, followed by a ripening phase that was delayed by a cold period, when the LWC decreases almost to 0 (Figure 4i). In the middle of May a runoff phase started. The backscattering
followed the different phases as expected. The season 2017/2018 is more regular, with a monotonic increasing of LWC indicating a short moistening followed by a regular ripening and the runoff. In this case the measured SWE anticipated the runoff onset of about one week w.r.t. the modeled SWE, which seems more in accordance with the S-1 data. The backscattering shows a similar behavior of other previously discussed cases with the characteristic "U-shaped" signal except for the T066 that present several oscillations in the VH polarization.

### 405   4.2   Temporal Evolution of the Backscattering

From the comparison carried out in the previous section and by taking into account the main backscattering mechanisms described in section 2.2, it is possible to derive and explain the temporal behavior of $\sigma^0$ generated by a C-band SAR over a sufficiently deep snowpack located in an open space that present a linear transition between the three melting phases. By analyzing the backscattering time series of the same pixel, the contribution of the LIA is always the same, making the values
of the time series comparable. Figure 5 shows an illustrative evolution of $\sigma^0$ for a complete hydrological year that summarizes both the state-of-the-art background and the observations done on real data. As described later, this conceptual time signature will allow to derive a set of rules for the identification of the melting phases also in time series of backscattering never observed before or in indipendent dataset (e.g., Veyssière et al. (2018); Lievens et al. (2019)).

Before the snow cover the terrain, $\sigma^0$ is influenced by the fluctuation of the soil moisture (Ulaby et al., 1996). Then, generally the first snow fall is wet or it covers relatively warm terrain resulting in a wet snowpack. This generates low backscattering values in the SAR response. This situation, which in alpine environments usually lasts for short periods, ends either with a significant decrease of the temperature that brings the snowpack to a dry condition or with a complete melting of the snowpack. It is also possible that the soil freezes before the first snowfalls. In this case the coefficient of backscattering decreases and stabilizes around a given value, not being affected by the soil moisture anymore.

As soon as the snowpack starts incorporating liquid water, the melting period starts. It can be divided into three important phases as described in section 2.1 i.e., the moistening, the ripening and the runoff phases. The first phase is related to the initial moistening of the snowpack. As discussed previously, the liquid water is introduced in the snow by rain and/or melt due to temperature and the incoming flux of shortwave radiation. At the beginning of the process the value of LWC is low and therefore the SAR backscattering experiences a relevant decrease in its value since the volumetric scattering dominates the total backscattering. The drop of the signal is recognizable by imposing a given threshold $T$. During the moistening, the wetting front may be visible only during the afternoon and not in the morning since the snowpack is still subjected to the diurnal cycles of melting and refreezing. As soon as the wetting front has penetrated the superficial insulating layer of the snowpack, the wet snow becomes visible also in the SAR early morning acquisitions. Please note that the systematic offset between the morning and afternoon signals represents the generally different local incidence angle of the ascending and descending acquisitions in mountainous region. At this point the phase of snowpack ripening starts. In this phase, the wetting front keeps penetrating the snowpack conducting it to an isothermal condition. During the ripening phase, which is influenced by the weather and the snowpack conditions, $\sigma^0$ varies according to the snow conditions but with an overall decreasing trend due to the increase of LWC.

We observed that the minimum of $\sigma^0$ is reached in correspondence of the finishing of the ripening phase and the beginning of the runoff phase for all the ten time series observed (see section 5). The runoff is instead characterized by a monotonic increase of the backscattering till all the snow is melted. To our knowledge, this characteristic behavior has been never observed in the literature before. Our interpretation is as follow: when the considered snowpack reaches its saturation condition in terms of LWC and snow structure, the backscattering recorded in C-band reaches its minimum value. This snowpack condition seems to correspond with the isothermal condition i.e., the end of the ripening phase. After the saturation point is reached, the monotonic increase of $\sigma^0$ could be explained by one or the combination of the following factors: i) an increase of the superficial roughness; ii) a change in the snow structure i.e., increase of the density and increase of grain size and; iii) at the end of the melting, the presence of patchy snow creates a situation of mixed contribution inside the resolution cell of the SAR and therefore a further increase of the total backscattering is recorded.

On the basis of this analysis, we propose here a simple set or rules to identify the snow melting phases on the basis of the multi-temporal SAR signal. The start of the melting process can be identified by a decrease of the multi-temporal SAR signal recorded in the afternoon of 2 dB or more w.r.t. the general winter trend. This threshold has been also proposed by (Nagler et al., 2016). As soon as also the backscattering time series recorded in the morning experience a decrease of more than 2 dB, we assume that the ripening phase begins. This phase, characterized by several oscillations, ends when both the morning and

afternoon $\sigma^0$ reach their local minimum. We propose the mean date among the local minima as the start of the runoff phase, which is characterized by a monotonic increase of the coefficient of backscattering. These rules are summarized in Algorithm 2. It is worth noting that, the rules are not calibrated on the observations done in section 4.1, but reflect the literature background.

---

**Algorithm 2:** Identification of the melting phases

**Input:** Multitemporal backscattering observations for different tracks, $\sigma_{morining}$ and $\sigma_{afternoon}$, for a given day

$\quad d$, $d \in \{1,..,d,..,D\}$ with $D$ total number of observations

**Output:** Onset moistening $T_M$, onset ripening $T_R$, onset runoff $T_{RO}$

**while** $d \leq D$ **do**

$\quad$ **if** $\sigma_{afternoon,d} - \sigma_{dry} \geq -2\ dB$ **then**

$\quad\quad$ # Snowpack is wet

$\quad\quad$ # **Check moistening phase**

$\quad\quad$ **if** $(\sigma_{morning,d} - \sigma_{dry} < -2\ dB)$ **then**

$\quad\quad\quad$ $T_M = d$

$\quad\quad\quad$ # **Do not check this condition anymore**

$\quad\quad\quad$ **continue**

$\quad\quad$ **end**

$\quad\quad$ # **Check ripening phase**

$\quad\quad$ **if** $(\sigma_{morning,d} - \sigma_{dry} \geq -2\ dB)$ **then**

$\quad\quad\quad$ $T_R = d$

$\quad\quad\quad$ # **Do not check this condition anymore**

$\quad\quad\quad$ **continue**

$\quad\quad$ **end**

$\quad\quad$ # **Check runoff phase**

$\quad\quad$ **if** $(\sigma_d == \sigma_{min})$ **then**

$\quad\quad\quad$ $T_{RO} = d$

$\quad\quad\quad$ # **Do not check this condition anymore**

$\quad\quad\quad$ **continue**

$\quad\quad$ **end**

$\quad$ **end**

$\quad$ **else**

$\quad\quad$ # Snowpack is dry

$\quad$ **end**

$\quad$ d ++

**end**

---

In the next section we applied these simple set of rules in order to identify the melting phases for each of the five considered test sites. Moreover, the same rules are used to identify the runoff onset for each SAR pixel in the topographically well-defined catchment of the Zugspitzplatt obtaining a spatially distributed map of the runoff timing.

## 5 Application of the proposed approach to 1D and 2D cases

In this section, we present the results obtained for the snow melting phases identification from the time series of backscattering recorded from S-1 over the five selected alpine test sites. The results are compared with the derivation of the melting phases considering the observed and modeled measurements of LWC and SWE. Finally, we present the result of the runoff onset identification in the two dimensional space of the original 20 meters SAR images for the Zugspitze catchment.

### 5.1 Identification of snow phases from Sentinel-1 in the five alpine test sites

Table 5 reports the comparison of the onset dates for the melting phases for each of the considered test sites. The phases were identified from the backscattering time series according to the rules expressed in the previous section. If more than two acquisitions i.e., ascending and descending are available for one test site, the first date representing the onset for the moistening and ripening phase among all available tracks is selected. The runoff onset is identified as the mean date among the local minima. These rules can be automatically applied without any human supervision.

On average, the moistening phase was identified with a r.m.s. error of 6.5 days. For the ripening phase the SAR time series allowed the identification with 4.5 days of r.m.s. error. Finally, the runoff was identified with a r.m.s. of 8 days (4 days r.m.s. error without considering Alpe del Tumolo for the years 2016/2017 and Weissfluhjoch for the years 2017/2018 where the runoff process were articulated). Considering the repetition frequency provided by S-1 and the possible uncertainty of the SNOWPACK modeling (Wever et al., 2015), the produced results demonstrate the effectiveness of using the SAR for characterizing the snow melt process.

In some cases, the proposed rules could not be applied and the onset could not be identified from the S-1 data. This is mainly due to the short melting or ripening periods that occurred during some years in the selected test sites. In these cases, the 6 days repetitions provided by S-1 is not adequate to sample this situation and it happens that the moistening phase is captured by the morning acquisition before than the afternoon acquisition (i.e., Zugspitze season 2016/2017 and 2017/2018, Clozner Loch season 2016/2017 second moistening phase and 2017/2018) or the first signal drop is reached at the same time of the local minima (i.e., Clozner Loch season 2017/2018). One can also notice that, for the first runoff identified in the season 2016/2017 for Malga Fadner, the proposed rules failed since for T168 no local minimum was clearly identified (Figure 4g).

### 5.2 Extension to a 2D analysis of the runoff onset: the Zugspitzplatt catchment

In this section we evaluate how the identification of the runoff onset is performed at a catchment scale. In particular, we considered the multi-temporal behavior of each pixel acquired by S-1 over the Zugspitzplatt during the hydrological year 2017/2018. The plateau (1500-2700 m a.s.l.) on the southern slope of Mt. Zugspitze summit (2962 m a.s.l.) is well suited for

this application scenario, since it is proven that all surface and ground water is drained to the Reintal valley in the east by the Partnach River (Rappl et al., 2010). With regard to a potential transport of contaminants that are stored in the snowpack and released with the first snowmelt (Hürkamp, Tafelmeier and Tschiersch, 2017), the knowledge of the runoff onset can provide important information for the scope of action concerning the management of countermeasures or planning actions to mitigate potential soil and water contamination.

As illustrated in the previous section, the runoff onset was identified by locating the minimum of the backscattering time series. In order to increase the robustness of the detection, we considered the mean of backscattering of close pixel presenting the same characteristics in terms of altitude, exposition and slopes. In detail, belts of 100 m were considered for the altitude. Slope was divided in three classes between 0-20, 20-40 and 40-60 degrees. Four aspect classes were considered, i.e. North, East, South and West. Finally, a local incidence angle ranging from 25 to 65 degrees was divided in 8 classes with 5 degrees span, avoiding layover and shadow effects. All the homogeneous classes generated by the different combinations were aggregated. The forested areas were masked using the Copernicus tree cover density map (https://land.copernicus.eu/pan-european/high-resolution-layers/forests/tree-cover-density/status-maps/2015). Moreover, since in this illustrative example we are interested in the main runoff contribution, the proposed algorithm is looking for local minima of the backscattering time series only after January 2018. This to exclude isolated wet snowfalls or complete early melting events typical of the beginning of the seasons.

Figure 6 shows the runoff onset identified by the proposed method. As one can notice, the regions at lower altitude started the runoff phase before the areas at higher altitude. The same consideration can be done for the pixels north exposed versus the south exposed ones. Interestingly, the last areas that start the runoff phase in the catchment are the glacierized areas (Northern and Southern Schneeferner glacier) and north faced slope areas. A selection of the backscattering time series is reported at the bottom of the Figure 6 for six points selected at different altitudes. As one can notice the characteristic behavior described in section 4.2 is always visible in the real data even though they were not analyzed before.

## 6  Discussion

Snow monitoring and/or prediction systems are typically based on real-time snow ground observations (e.g., WSL Swiss monitoring system https//www.slf.ch/en/avalanche-bulletin-and-snow-situation/snow-maps.html), snow hydrological models (e.g., Mysnowmap for the European Alps https//www.mysnowmaps.com/), optical and passive microwave remote sensing observations (e.g., ESA Climate Change snow Initiative snow-CCI http//cci.esa.int/snow), or the combination of different sources (e.g., the US National 515 Operational Hydrologic Remote Sensing Center (NOHRSC) https//www.nohrsc.noaa.gov/). The accuracy of such systems varies, but in general is limited by the poor information on snow precipitation, especially in mountain areas. This could lead to errors of several days, even weeks, in the estimation of the snow disappearance time (Engel et al., 2017). The approach described in this paper allowed the identification of the melting phases for the five considered test sites with an rmse of 6 days for the 510 moistening phase, 4 days for the ripening and 7 days for the runoff phase. Therefore, it could be potentially useful to improve the performances of snow monitoring.

It is important to underline that, in order to predict runoff, further hydrological modeling is needed beside the information provided by the proposed approach. While the runoff production below the snowpack starts quickly, being snow permeable to water, then the streamflow production can be delayed of several days, even weeks, depending on catchment size and hydrological behavior (Rinaldo et al., 2011). Therefore, even if we do not propose a real-time implementation, we think that, combining the information on the snow melting phases based on the principles presented in Section 4.2 and easily available real-time and historical auxiliary data such as temperature or historical streamflow, it is possible to develop an algorithm to extract valuable information for the anticipation of the peak stream runoff phase.

Knowing the snow melting phases with just a few days delay can have very important applications for water resources management (e.g., hydropower production or irrigation administration). In detail, the information provided by the proposed approach can be ingested in operational hydrological modeling systems. In detail, the ingestion of remote sensing information for improving snow modeling and monitoring has been extensively applied in the past e.g., (Molotch and Margulis, 2008). So far, the most common variable assimilated is snow cover fraction from optical sensors since this is the most available information acquired using remote sensing. In our case, we would need to assimilate either information on presence/absence of snow liquid water content or on the snow depletion curve, which can be computed for the first time from the real beginning of the melting (i.e., runoff onset) from high resolution remote sensing data. From a theoretical point of view, this is feasible. However, if the assimilated variable is snow liquid water content, only snow models which explicitly simulate snow liquid water content can be used. Usually physically based, energy-based snow models such as GEOtop (Endrizzi et al., 2014), AMUNDSEN (Strasser et al., 2011), CROCUS (Brun et al., 1992) or SNOWPACK/ALPINE3D (Bartelt and Lehning, 2002; Lehning et al., 2006) are suitable for this purpose.

The possibility to use state of the art Radiative Transfer (RT) models to simulate the multitemporal behavior of the backscattering presented in Section 4.2 has also been investigated. Although wet snow is of great importance for many applications, the most widely used models have been tested and applied mainly in dry snow conditions (Picard et al., 2018; Proksch et al., 2015). In detail, during the melting process the increase of superficial roughness, LWC and density and the coarsening of the snow grains play an important role on the backscattering mechanisms. Indeed, when the LWC increases, the absorption coefficient increases, the penetration depth decreases, and the total backscattering is influenced more and more by the superficial roughness of the snow. As discussed in the background section 2.2, at the best of our knowledge, only few works have specifically addressed the wet snow modeling at C-band i.e., (Shi and Dozier, 1995; Nagler and Rott, 2000; Magagi and Bernier, 2003). Differently from more advanced models such as SMRT (Picard et al., 2018) or MEMLS3&a (Proksch et al., 2015), these models assume independent scattering. Even though Shi and Dozier (1995) and Magagi and Bernier (2003) indicate a positive correlation between largely wet snowpack and the superficial roughness, Kendra et al. (1998) on the basis of ground experimental analysis, expressed some doubts on the realistic behavior of such models. Therefore, wetsnow RT modeling requires dedicated efforts and validation campaigns, which has never been systematically conducted for characterizing the multi-temporal snow roughness, which are out of the scope of this paper and will be left as future work.

It is finally worth noting that the availability of multi-temporal data, acquired regularly over the entire globe and freely accessible, opens new opportunities to monitor dynamic phenomena. In particular, monitor snow depth and snow water equiv-

alent in a systematic and spatially distributed manner would be crucial for a proactive management of the water resources. The recent paper by Lievens et al. (2019) proposes an empirical algorithm for snow depth retrieval from S-1 at 1 km resolution. The authors suggest a C-band sensibility to snow height generated by the cross-polarized information. This was never fully recognized before in the literature. Even though the focus of our research is only on the snowmelt, by considering the 20 meters multi-temporal S-1 data acquired over the five test sites studied in the presented work, we provide some remarks that may be useful for future works in this context. If, from one side all the backscattering time series in the two polarization showed in the paper by Lievens et al. (2019) exhibit the characteristic shape identified and analyzed in the presented study, on the other hand, at least in our five test sites – which is a restricted and very specific dataset w.r.t. the global one considered by Lievens et al. – the ratio $\sigma_{VH}/\sigma_{VV}$ seems not providing clear evidences that the cross polarization is sensible to the increase (or decrease) of snow depth (or SWE) during both the accumulation and melting period (see Figure 4). However, this does not exclude that different manipulation of the S-1 data (e.g., spatial and temporal averaging) and the empirical incidence angle normalization proposed in Lievens et al. (2019), which were not taken into account in our experiments, may contribute to increase the sensitivity of the backscattering to the snow height, by possibly removing the source of noise. In conclusion, despite the lack of a generally accepted physical explanation, this work shows how the rich amount of SAR data made available with a high repetition interval can allow the monitoring of the complex processes related to the snow evolution in a manner that was never addressed before. We believe this will be one of the most interesting research topics in the future.

## 7 Conclusions

In this paper, we analyzed the correlation between the multi-temporal SAR backscattering and the snow melt dynamics. We compared Sentinel-1 backscattering with LWC and SWE measurements derived from in situ observations and process-based snow modeling simulations for five alpine test sites in Italy, Germany and Switzerland considering two hydrological years. We found that the multi-temporal SAR measurements allow the identification of the three melting phases that characterize the melting process i.e., moistening, ripening and runoff with a good agreement considering the revisit time of Sentinel-1. In detail, we found that in the considered sites the SAR backscattering decreases as soon as the snow starts containing water, and that the backscattering increases as soon as SWE starts decreasing, which corresponds to the release of meltwater from the snowpack. We discuss the possible reasons of this increase, which are not directly correlated to the SWE decrease, but most probably to the different snow conditions, which change the backscattering mechanisms. From this study we define a set of simple rules that can be applied to the multi-temporal SAR backscattering in order to identify the melting phases. We showed that by applying these rules, the identification of the melting phases was possible for the five considered test sites with an rmse of 6 days for the moistening phase, 4 days for the ripening and 7 days for the runoff phase. Moreover, the same rules were applied for the identification of the runoff onset for the entire Zugspitzplatt catchment with reasonable results even if further hydrological analyses have to be performed. The presented investigation could have relevant application for monitoring and predicting the snowmelt progress over large regions. A better understanding of the spatial and temporal evolution of melting

dynamics in mountain regions and the knowledge on the onset of melt water runoff can help to predict floods and define the scope of action to mitigate potential contaminant distributions in soils and surface water.

As future developments we plan to develop and test an automatic method to identify the three melting phases of a snowpack using larger validation dataset (e.g., SNOTEL) and allow to proper discuss the spatial and temporal evolution of snow water content and runoff in mountainous region. Moreover, we investigate the reasons of the increase of the backscattering in corre-

590 spondence of the decrease of SWE through in situ experiments that take into account the hypothesis expressed in this paper. This will help the development of the RT models in wet snow conditions.

*Data availability.* Relevant data can be made available upon request to the authors. All the Sentinel-1 data are freely available at https://scihub.copernicus.eu/ upon registration.

*Author contributions.* CM, MC and GB designed the research; VP carried out all the experiments and run the SNOWPACK model; all the

595 authors contributed to the analysis and interpretation of the results; CM wrote the paper based on inputs and feedbacks from all co-authors.

*Competing interests.* Authors declare no competing interests.

*Acknowledgements.* We thank the Bavarian Avalanche Warning Service (Lawinenwarnzentrale Bayern) and the Environmental Research Station Schneefernerhaus (UFS) for providing the measurement data for the German test site Zugspitze, the Hydrografic office of the Autonomous province of Bolzano for providing the data for the Italian test sites of Alpe Tumulo, Clozner Loch and Fadner Alm and the WSL

insitute for snow and avalanche research SLF for providing the data for the Switzerland test site Weissfluhjoch. The work of the MC and GB were financed through the CRYOMON-SciPro project, founded by the Euregio Science Fund 1st call, project number IPN 10. Parts of the measurements and snow sensor installations at the Zugspitze station were funded by the Bavarian Ministry of the Environment and Consumer Protection (BayStMUV) in the framework of the Virtual Alpine Observatory (VAO) project.

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

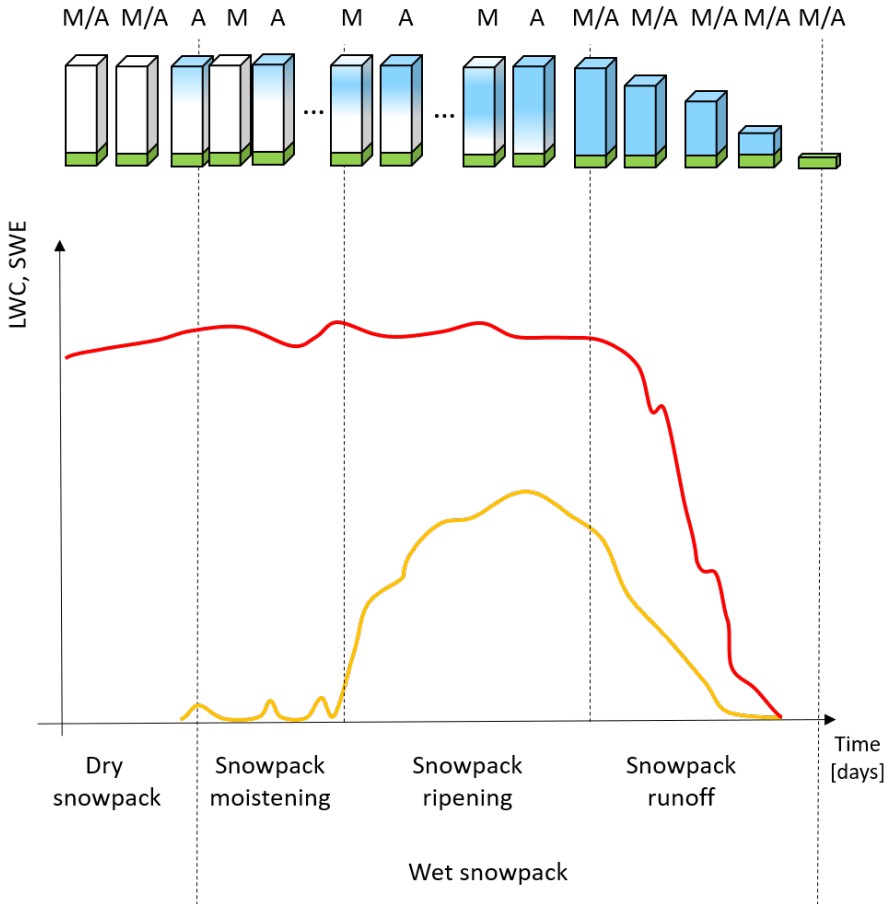

**Figure 1.** Example of transitions in snow status during the melting season obtained by sampling the snow in the morning (M), when the S-1 descending observations are taken, and in the evening (A) when the S-1 ascending data are taken. The upper part of the figure illustrates the simplified temporal transportation of the free liquid water (blue area) in the dry snowpack (white area). The lower part of the figure illustrates the respective temporal evolution of LWC (yellow line) and SWE (red line). In detail, by starting from a dry situation, the liquid water is introduced into the snowpack by either a rain event or the melt due to the incoming flux of shortwave radiation. In this moistening phase the LWC (yellow line) varies with a diurnal cycle. Repeated cycles of partial melting and refreezing conduce the snowpack to the isothermal state. During the ripening period, a combination of different situations can occur depending on the weather conditions but an increasing trend of the LWC is visible. Once the snowpack is isothermal and it cannot retain water anymore, it starts to produce water output until it melts totally. This last phase starts with a significant decrease of the SWE (red line).

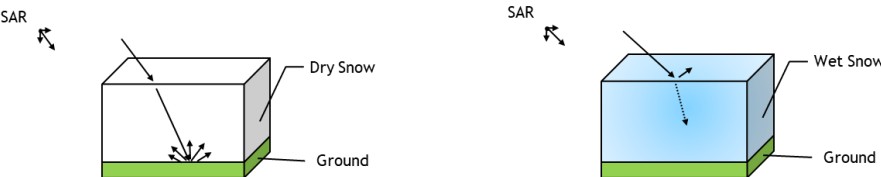

**Figure 2.** Main SAR backscattering mechanisms in presence of dry and wet snow at C-band. The dry snow is almost transparent, and the radar echo can penetrate the snow for several meters. The presence of LWC, causes high dielectric loss, which increases the absorption coefficient.

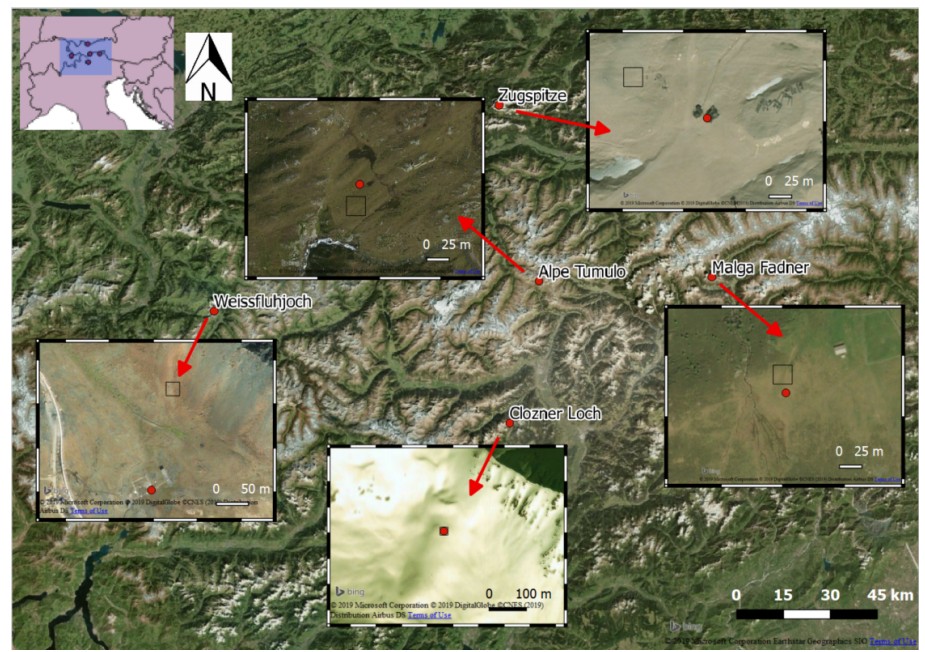

**Figure 3.** Overview map with the five stations used for the presented study (©2019 Microsoft Corporation ©2019 Digital Globe ©CNES(2019) Distribution Airbus DS). The red points indicate the exact location of the stations. The black squares indicate the S-1 footprints. The footprints were selected in order to minimize any possible interference of the EM wave with the homemade structures but maintaining a certain correlation with the in situ measurements. The panoramic images give an idea about the land cover type and the topography around the stations.

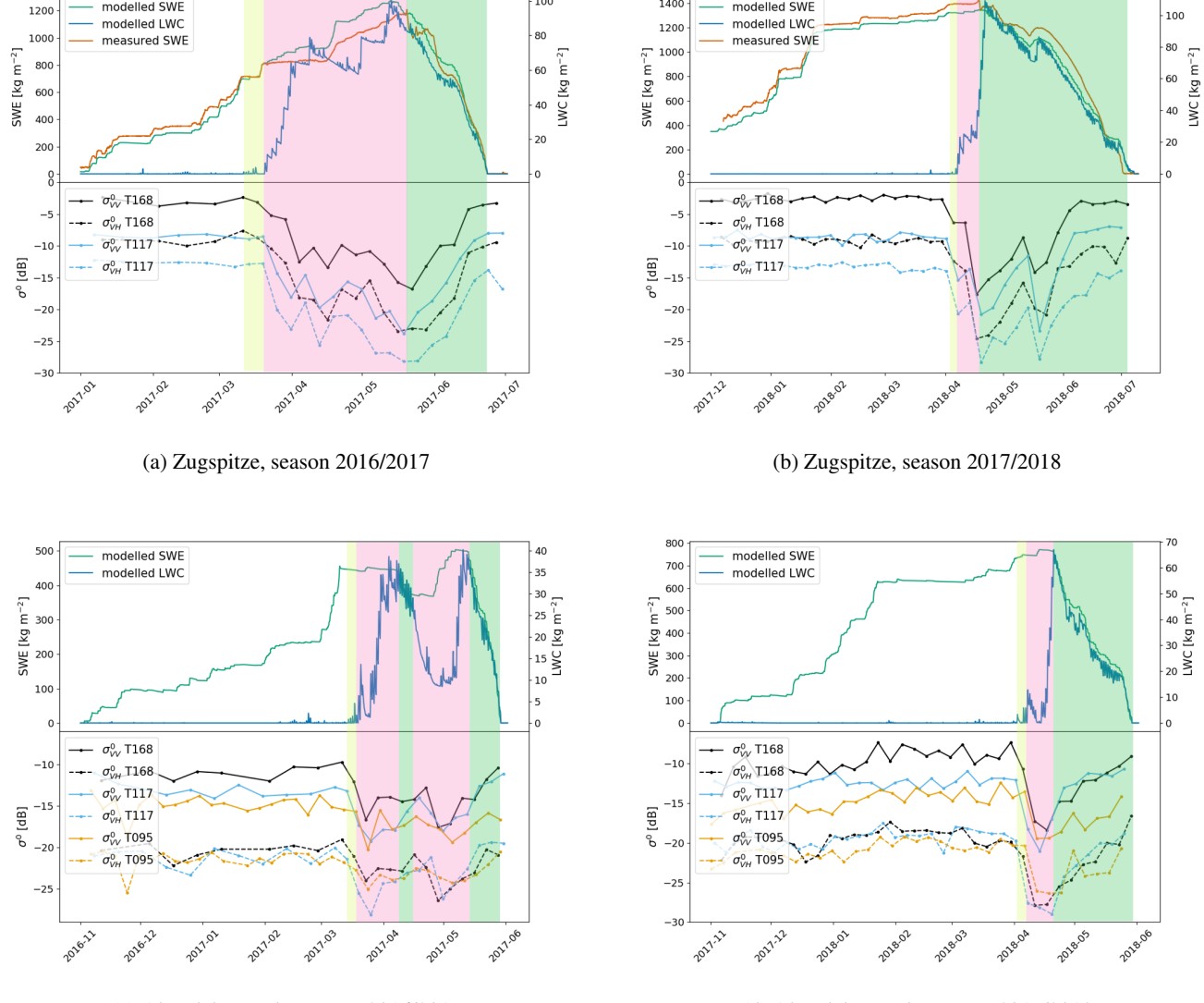

(a) Zugspitze, season 2016/2017

(b) Zugspitze, season 2017/2018

(c) Alpe del Tumulo, season 2016/2017

(d) Alpe del Tumulo, season 2017/2018

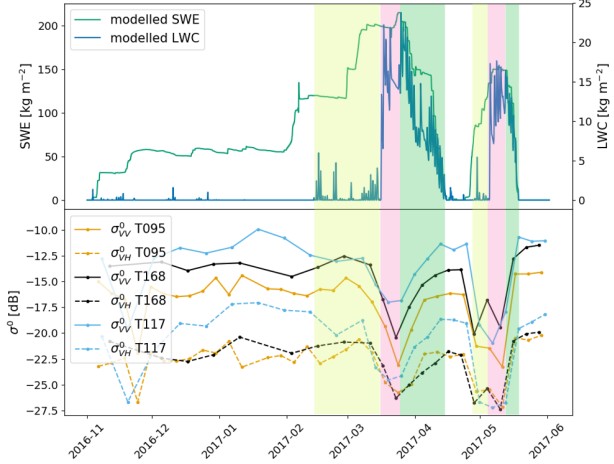

(e) Clozner Loch, season 2016/2017

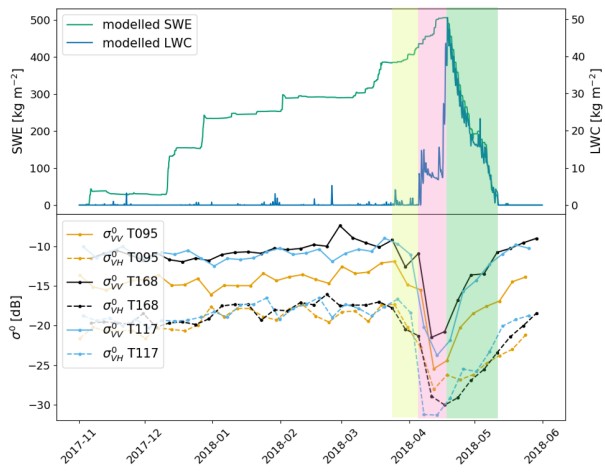

(f) Clozner Loch, season 2017/2018

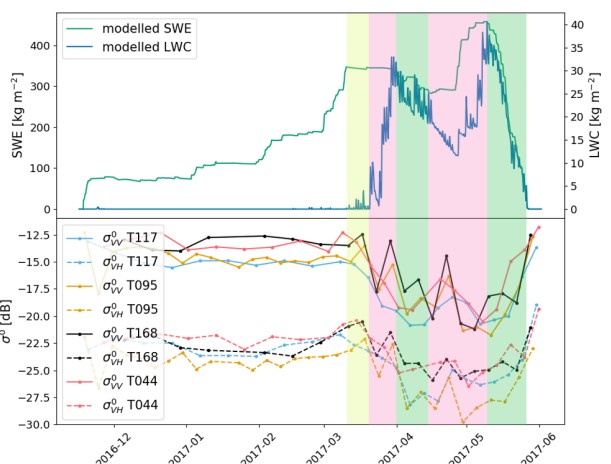

(g) Malga Fadner, season 2016/2017

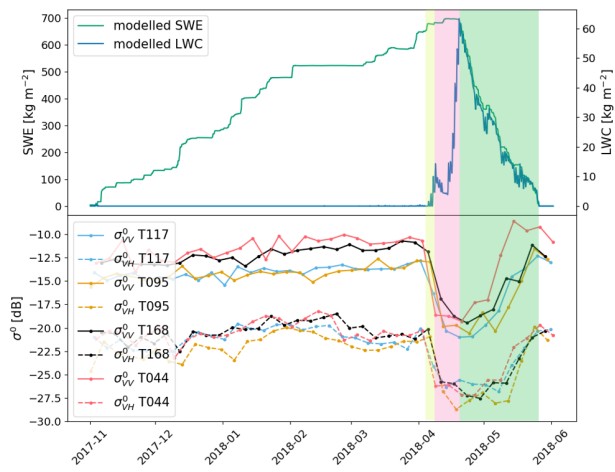

(h) Malga Fadner, season 2017/2018

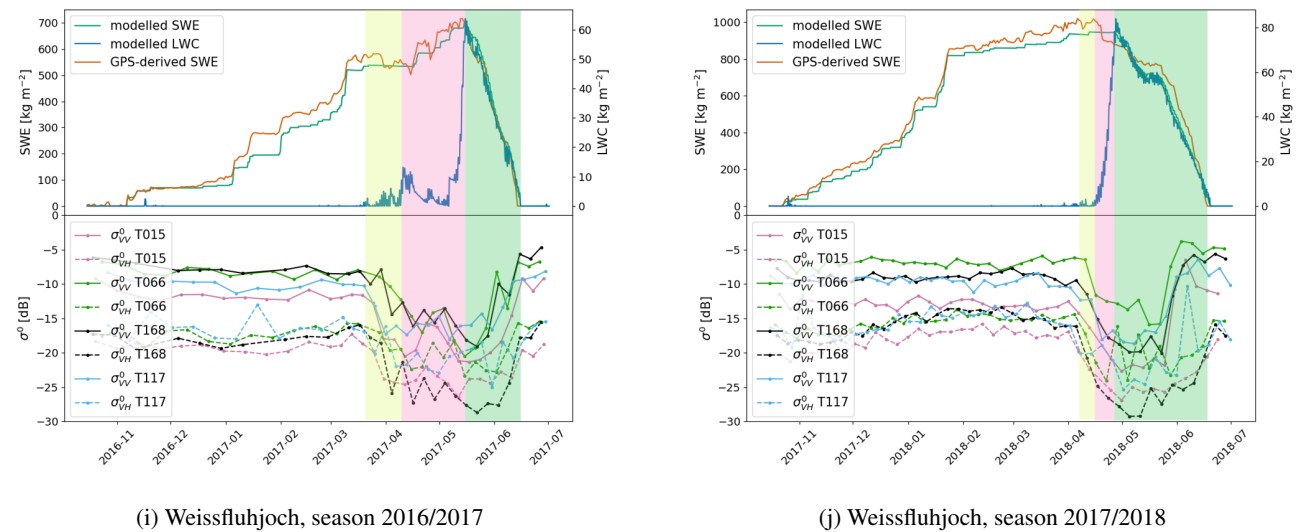

(i) Weissfluhjoch, season 2016/2017          (j) Weissfluhjoch, season 2017/2018

**Figure 4.** Temporal evolution of the coefficient of backscattering acquired over the five test sites compared to LWC and SWE measured in situ at the stations (when available) and modeled with SNOWPACK (contains modified Copernicus Sentinel data, 2016/2018, processed by Eurac Research). The three phases during the melting have been identified from the in situ/modeled data. The first phase of moistening is reported in light yellow, the ripening phase in light red and the runoff in light green. For all the test sites we found that the multi-temporal SAR measurements confirm the identification of the three melting phases. In detail, we systematically found that the SAR backscattering decreases as soon the snow starts containing water and increases as soon as SWE starts decreasing, which corresponds to the release of meltwater from the snowpack.

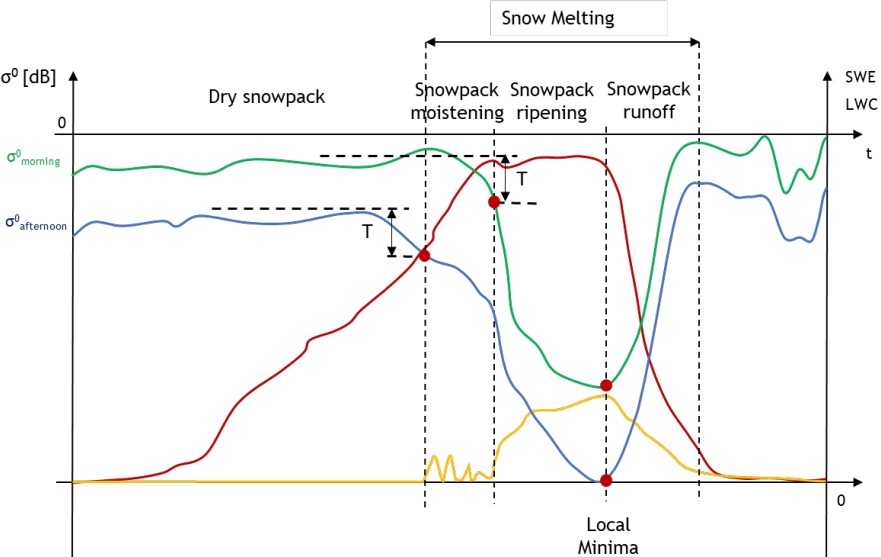

**Figure 5.** Schematic representation of the evolution of the backscattering coefficient acquired in the morning (green line) and in the afternoon (blue line) compared with LWC (yellow line) and SWE (red line) evolution. The offset between the morning and afternoon signals is due to the generally different local incidence angle of the ascending and descending acquisitions in mountainous regions. The three melting phases are identified from the LWC and SWE information. Correspondingly, the rules for the identification of each phase from the time series of $\sigma^0$ is highlighted: a decreases of at least T [dB] from the mean value in dry snow condition applied to the afternoon and morning signals identifies the moistening and ripening onsets respectively. The local minima of the signals indicate the runoff onset.

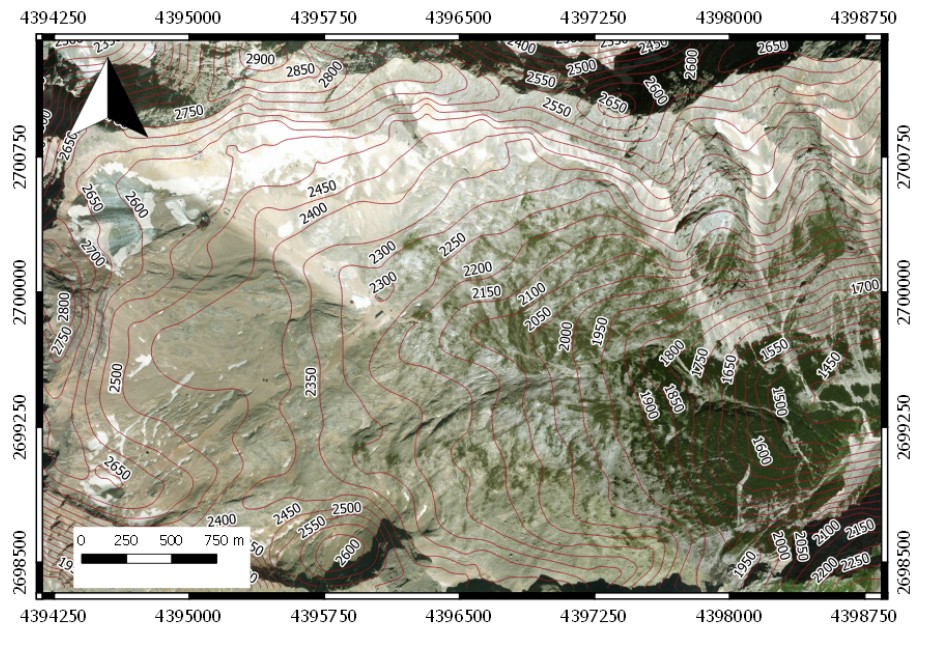

(a)

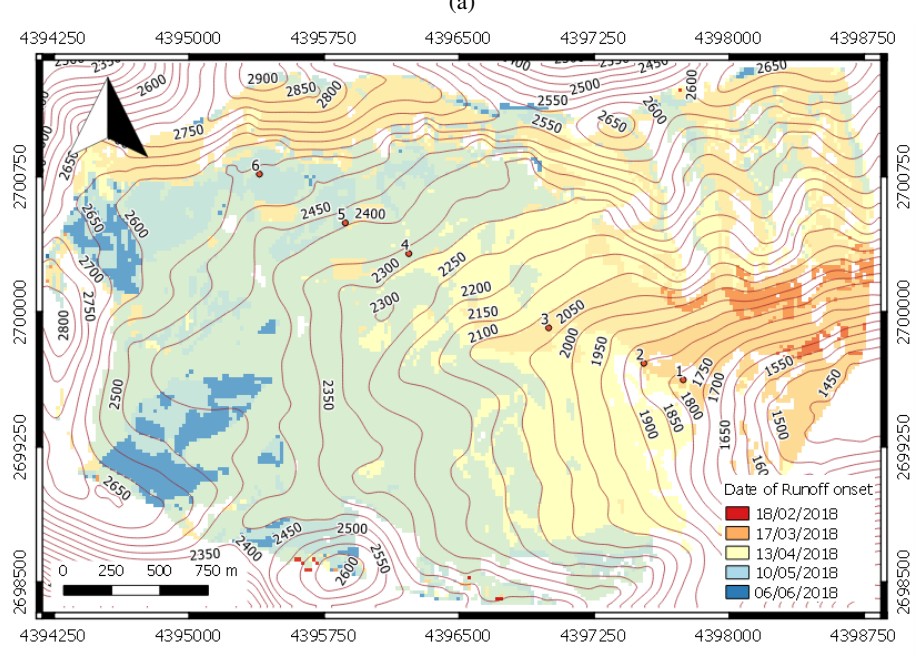

(b)

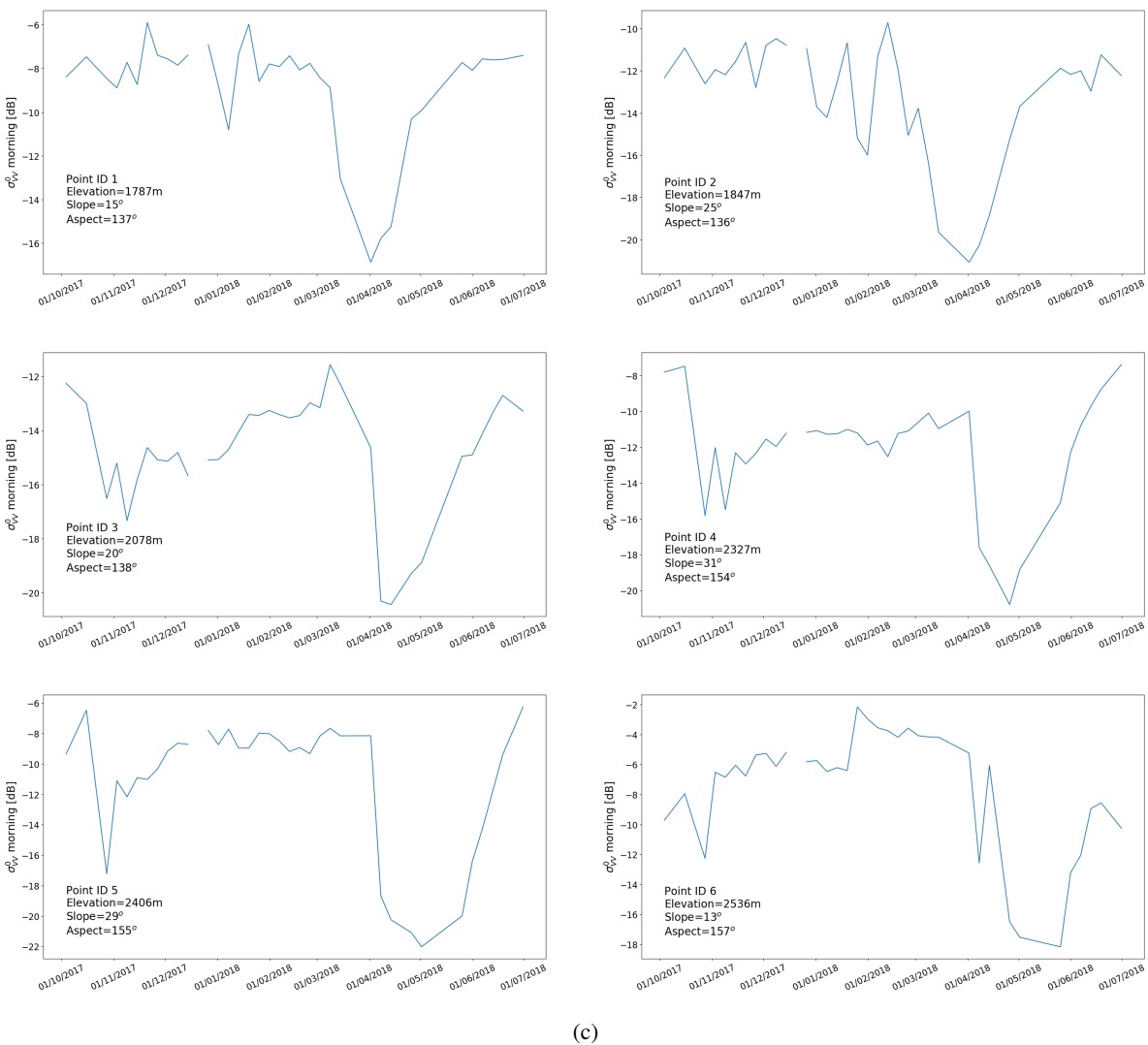

(c)

**Figure 6.** Runoff onset for the Zugspitzplatt catchment. (a) Test site presentation (©2019 Microsoft Corporation ©2019 Digital Globe ©CNES(2019) Distribution Airbus DS) (b) Map of the runoff onset (contains modified Copernicus Sentinel data, 2018, processed by Eurac Research). The runoff started at lower altitude and at the south exposed slopes. The last areas to have the runoff in the catchment are the high-altitude area, the north exposed and glacierized areas. (c) The multitemporal backscattering time series for the selected points identified in (b). All the time series present the characteristic "U-shaped" pattern.

**Table 1.** Simplified SAR backscattering response to wet snow divided in volumetric, $\sigma_{vol}^0$, and surface backscattering, $\sigma_{sup}^0$, contributions. Considering a sufficiently thick snowpack the contribution of $\sigma_{grd}^0$ can be negleted.

| Parameter | $\sigma_{vol}^0$ | $\sigma_{sup}^0$ |
|---|---|---|
| Liquid water content (LWC) | negative correlation | positive correlation |
| Snow density (DS) | negative correlation | positive correlation |
| Snow grain size (GS) | positive correlation | - |
| Surface roughness (RS) | - | positive correlation |

**Table 2.** Details of the meteorological and snow parameters measured at each station. Wind velocity (VW), wind direction (DW), air temperature (TA), relative humidity (RH), snow depth (HS), snow temperature at different depths (TS), surface temperature (TSS), soil temperature (TSG), incoming shortwave radiation (ISWR), incoming longwave radiation (ILWR), outgoing shortwave radiation (OSWR), snow water equivalent (SWE), snow density (DS), liquid water content (LWC) and ice content (IC).

| Station | Latitude, Longitude | Altitude [m a.s.l.] | Available measurements |
|---|---|---|---|
| Zugspitze (Germany) | 10.9835, 47.4064 | 2420 | VW, DW, TA, RH, HS, TSS, ISWR, OSWR, SWE, DS, LWC, IC |
| Alpe del Tumulo (Italy) | 11.1487, 46.9136 | 2230 | VW, DW, TA, RH, HS, TS, TSS, TSG, ISWR |
| Clozner Loch (Italy) | 11.0283, 46.5134 | 2165 | VW, DW, TA, RH, HS, TS, TSS, TSG, ISWR |
| Malga Fadner (Italy) | 11.8614, 46.9256 | 2155 | VW, DW, TA, RH, HS, TS, TSS, TSG, ISWR |
| Weissfluhjoch (Switzerland) | 9.8096, 46.8296 | 2455 | VW, DW, TA, RH, HS, TSS , TSG, ISWR, OSWR, SWE |

**Table 3.** SNOWPACK calibration results for each test site. Pearson correlation coefficient ($\rho$) and the mean absolute error (MAE) have been computed for snow depth (HS), snow temperatures at three different depth TS1 (0 m from the ground), TS2 (0.2 m from the ground), TS3 (0.5 m from the ground) and SWE, according to the availability of the ins situ data.

| Station | Roughness [m] | Calibration results | | | | | | | | | |
|---|---|---|---|---|---|---|---|---|---|---|---|
| | | HS | | TS1 | | TS2 | | TS3 | | SWE | |
| | | $\rho$ | MAE [cm] | $\rho$ | MAE [°C] | $\rho$ | MAE [°C] | $\rho$ | MAE [°C] | $\rho$ | MAE [kgm$^{-2}$] |
| Zugspitze | 0.005 | 0.99 | 3.7 | - | - | - | - | - | - | 0.99 | 47.8 |
| Alpe Tumulo | 0.03 | 0.99 | 3.6 | 0.90 | 0.4 | 0.93 | 0.4 | 0.88 | 0.5 | - | - |
| Clozner Loch | 0.01 | 0.99 | 4.1 | 0.87 | 0.8 | 0.78 | 1.8 | - | - | - | - |
| Malga Fadner | 0.01 | 0.99 | 2.8 | 0.83 | 0.6 | 0.83 | 0.7 | 0.85 | 1.2 | - | - |
| Weissfluhjoch | 0.002 | 0.99 | 2.8 | - | - | - | - | - | - | 0.99 | 35.1 |

**Table 4.** List of the Sentinel-1 acquisitions and their main characteristics over the five test sites.

| Test Site | Relative orbit number (i.e., track number) | Time of the acquisition | Orbit Direction | Local incidence angle (LIA) |
|---|---|---|---|---|
| Zugspitze | 117 | Afternoon | Ascending | 38° |
|  | 168 | Morning | Descending | 39° |
| Alpe Tumulo | 095 | Morning | Descending | 47° |
|  | 117 | Afternoon | Ascending | 35° |
|  | 168 | Morning | Descending | 40° |
| Clozner Loch | 095 | Morning | Descending | 43° |
|  | 117 | Afternoon | Ascending | 39° |
|  | 168 | Morning | Descending | 36° |
| Malga Fadner | 044 | Afternoon | Ascending | 34° |
|  | 095 | Morning | Descending | 48° |
|  | 117 | Afternoon | Ascending | 46° |
|  | 168 | Morning | Descending | 38° |
| Weissfluhjoch | 015 | Afternoon | Ascending | 43° |
|  | 066 | Morning | Descending | 31° |
|  | 117 | Afternoon | Ascending | 33° |
|  | 168 | Morning | Descending | 41° |

|  | S-1 | Reference | Difference [days] |
|---|---|---|---|
| Moistening | - | 11/03/2017 | - |
| Ripening | 23/03/2017 | 21/03/2017 | +2 |
| Runoff | 20/05/2017 | 20/05/2017 | 0 |

(a) Zugspitze, season 2016/2017

|  | S-1 | Reference | Difference [days] |
|---|---|---|---|
| Moistening | - | 04/04/2018 | - |
| Ripening | 05/04/2018 | 08/04/2018 | -3 |
| Runoff | 18/04/2018 | 18/04/2018 | 0 |

(b) Zugspitze, season 2017/2018

|  | S-1 | Reference | Difference [days] |
|---|---|---|---|
| Moistening | 19/03/2017 | 14/03/2017 | +5 |
| Ripening | 23/03/2017 | 20/03/2017 | +3 |
| Runoff 1 | 24/03/2017 | 08/04/2017 | -14 |
| Runoff 1 | 01/05/2017 | 13/05/2017 | -13 |

(c) Alpe del Tumulo, season 2016/2017

|  | S-1 | Reference | Difference [days] |
|---|---|---|---|
| Moistening | 07/04/2018 | 02/04/2018 | +5 |
| Ripening | 11/04/2018 | 07/04/2018 | +4 |
| Runoff | 14/04/2018 | 20/04/2018 | -6 |

(d) Alpe del Tumulo, season 2017/2018

|  | S-1 | Reference | Difference [days] |
|---|---|---|---|
| Moistening 1 | 23/02/2017 | 14/02/2017 | +9 |
| Moistening 2 | - | 29/04/2017 | - |
| Ripening 1 | 12/03/2017 | 16/03/2017 | -4 |
| Ripening 2 | 28/04/2017 | 05/05/2017 | -7 |
| Runoff 1 | 22/03/2017 | 25/03/2017 | -3 |
| Runoff 2 | 08/05/2017 | 13/05/2017 | -5 |

(e) Clozner Loch, season 2016/2017

|  | S-1 | Reference | Difference [days] |
|---|---|---|---|
| Moistening | - | 25/03/2018 | - |
| Ripening | - | 06/04/2018 | - |
| Runoff | 12/04/2018 | 18/04/2018 | -6 |

(f) Clozner Loch, season 2017/2018

|  | S-1 | Reference | Difference [days] |
|---|---|---|---|
| Moistening | 19/03/2017 | 14/03/2017 | +5 |
| Ripening | 23/03/2017 | 20/03/2017 | +3 |
| Runoff 1 | 10/04/2017 | 30/03/2017 | +11 |
| Runoff 2 | 07/05/2017 | 09/05/2017 | -2 |

(g) Malga Fadner, season 2016/2017

|  | S-1 | Reference | Difference [days] |
|---|---|---|---|
| Moistening | 07/04/2018 | 05/04/2018 | +2 |
| Ripening | 11/04/2018 | 07/04/2018 | +4 |
| Runoff | 21/04/2018 | 19/04/2018 | +2 |

(h) Malga Fadner, season 2017/2018

|  | S-1 | Reference | Difference [days] |
|---|---|---|---|
| Moistening | 25/03/2017 | 19/03/2017 | +6 |
| Ripening | 04/04/2017 | 09/04/2017 | -5 |
| Runoff | 14/05/2017 | 16/05/2017 | -2 |

(i) Weissfluhjoch, season 2016/2017

|  | S-1 | Reference | Difference [days] |
|---|---|---|---|
| Moistening | 06/04/2018 | 02/04/2018 | +4 |
| Ripening | 10/04/2018 | 17/04/2018 | -7 |
| Runoff | 08/05/2018 | 19/04/2018 | +19 |

(j) Weissfluhjoch, season 2017/2018

**Table 5.** Onset times for the melt phases identified in the five test sites using the LWC and SWE (reference) and Sentinel-1 with the method proposed in the previous section.