# Peer review of "Use of Sentinel-1 radar observations to evaluate snowmelt dynamics in alpine regions"

_The Cryosphere, 2019_

## Referee Comment (RC1) · Anonymous Referee #1 · 8 Oct 2019

General comments:

This paper tries to link the backscattered signal of C-Band SAR Sentinel-1 data to the main 3 melt periods in alpine regions: moistening, ripening and runoff. This work is also supported by physical snow modeling using SNOWPACK and in-situ dataset from 5 different monitoring stations in the Alps.

I really appreciate the physics based explanation of $\sigma^0$ variations. That being said with the information in this manuscript, it is really not clear to me how the authors generated the "theoretical" curves of figure 5. More explanation and details on how the authors generated those curves is needed. What input data was used?

Another important factor which might be linked to the previous comment is that it seems

the authors used the behaviors observed at the fives sites to describe the theoretical curves generated and used to create their approach to detect the 3 main melt phases. The authors then use those same sites to validate the approach which I find redundant. The authors would need independent data to validate the approach. This would not be needed if the approach was based on theory of $\sigma^0$ behaviors and those behaviors are observed in the S-1 data. Again, I feel like the curves generated in figure 5 need better explanation.

Nonetheless, the authors seem to understand the different radiative transfer interaction of the microwave signal with the different snowpack properties. It would have been nice to see some radiative transfer modeling from SMRT or a similar RT model to simulate the $\sigma^0$ behavior.

Specific comments:

L.5. Remove "be" in "to be obtained".

L.51. Change "The establishing"

L.76. change "has demonstrated" to "was shown"

L.94-95. This nominal resolution is only true for the high res IW mode. It can be removed in this section of the text since it is better described in the data section.

L.98. Remote "the" in "the monitoring"

L.106. change "polarimetric" to "polarization"

L.144. correct "properties"

L.171. remove "round the grains", metamophism does not always round the grains, more complex shapes can be created. Simply remove this part.

L.187. change to "October 1, 2016"

L.280. correct "removal"

L.411-412. not clear to me what you mean by depolarization here. To me depolarized signal implies that the V transmit is switched to H thus increasing VH and decreasing VV. An ice layer alone would not depolarize your signal, it would affect the scattering by adding a reflective layer in your snowpack.

---

## Referee Comment (RC2) · Anonymous Referee #2 · 18 Nov 2019

This is a well-written and clearly organized paper which utilizes the rich new time series of C-band SAR data from Sentinel-1 to explore the extent to which remote sensing can provide snow melt state information for Alpine snowpacks. Detailed analysis of simulations from the physical model SNOWPACK at sites with comprehensive snow state and meteorological measurements allows extension of the approach broader regions. This provides a realistic approach with respect to operational implementation. I have a number of comments which will hopefully constructively improve the final manuscript.

1. Line 143: to what extent are C-band measurements influenced by snow grain size/shape?

2. Line 289: "These phases have been identified from the SWE and LWC data according to section 2.1." Were quantitative threshold values of LWC or change in SWE used

to identify the three different phases? If so, these are not described in Section 2.1. Some additional detail on how the three melt phases were classified based on the in situ snow measurements would be helpful.

3. Section 4.1 provides a detailed description of the S-1 backscatter time series as they relate to snow observations and SNOWPACK simulations. In general, the text provides sufficient explanations for what is shown in Figure 4. This provides a clear observational basis for the synthesis in Section 4.2. My main concern is then the statement on line 385 that "Figure 5 shows the theoretical temporal evolution of backscatter for a complete hydrological year." While the conceptual framework of this figure is based on the measurements in Figure 4, the 'theoretical' component of this figure would be stronger if it contained actual backscatter simulations using a radiometric model (there are numerous options but SMRT comes to mind as a logical choice). I suggest the schematic approach to Figure 5 be augmented with radiometric simulations.

4. The Conclusion section is quite brief, and does not include a discussion in a number of relevant areas. A couple of suggestions to expand this section: >There is recent work which suggests SWE can be retrieved from cross-pol C-band SAR measurements, including in the Alps. I agree with your statement on line 150 that "During the accumulation period, dry snow is almost transparent for C-band..." C-band sensitivity to SWE defies a physical explanation. The time series of Sentinel-1 data in your study provide no evidence of sensitivity to SWE (e.g. Figure 4). Can you provide a comment on this in the Discussion, in the context of the work by Lievens et al? https://doi.org/10.1038/s41467-019-12566-y

>Are the accuracies produced from this study (expressed as the RMSE's on line 495) sufficient to improve current prediction systems used in the Alps? Are operational hydrological modeling systems ready to implement the ingestion of snow state estimates from C-band SAR data or are there any impediments? Since the technique relies on the timing of the backscatter minima to identify the change from ripening to runoff, what is the latency in which a Sentinel-1 derived runoff retrieval could be used given the repeat interval of 6 days (for example, do you need to wait 6 times x number of days to ensure the minima was reached?)?

Editorial: Line 30 – The meaning of the sentence starting with "Precise predictions of the timing..." is not clear. I suggest removing it - the following sentence is more impactful anyway.

Line 36: I don't think 'favored by' is the right word choice...'driven by'?

Line 46: sentence starting "An increase of LWC..." – this is a very long sentence with many commas. Split into two sentences for readability.

Line 49: change to "Continuous measurements of SWE..."

Line 502 – if this is in reference to the western United States network of sites, it should be noted as 'SnoTel'.

Looks like the panels of Figure 4 are separated by Table 1 and Table 2?

Figure 6b requires a legend to indicate the values of the colour scale.

---

## Author Comment (AC1) · 20 Dec 2019

**Answer to the Anonymous Referee #1 – Manuscript tc-2019-175**

*This paper tries to link the backscattered signal of C-Band SAR Sentinel-1 data to the main 3 melt periods in alpine regions: moistening, ripening and runoff. This work is also supported by physical snow modeling using SNOWPACK and in-situ dataset from5 different monitoring stations in the Alps. I really appreciate the physics based explanation of $\sigma^0$ variations.*

We thank the anonymous Reviewer for his/her positive comments.

*That being said with the information in this manuscript, it is really not clear to me how the authors generated the "theoretical" curves of figure 5. More explanation and details on how the authors generated those curves is needed. What input data was used?*

*Another important factor which might be linked to the previous comment is that it seems the authors used the behaviors observed at the fives sites to describe the theoretical curves generated and used to create their approach to detect the 3 main melt phases. The authors then use those same sites to validate the approach, which I find redundant. The authors would need independent data to validate the approach. This would not be needed if the approach was based on theory of $\sigma^0$ behaviors and those behaviors are observed in the S-1 data. Again, I feel like the curves generated in figure 5 need better explanation. Nonetheless, the authors seem to understand the different radiative transfer interaction of the microwave signal with the different snowpack properties.*

*It would have been nice to see some radiative transfer modeling from SMRT or a similar RT model to simulate the $\sigma^0$ behavior.*

We are thankful to the Reviewer to point out this critical issue allowing us to better clarify it. Figure 5 is an illustrative representation of the phenomenological relationship between the multi-temporal backscattering and the snow evolution during a hydrological year. It should not be considered as a "theoretical" curve, but more as a conceptual scheme to better illustrate our interpretation of the backscattering signal in the sites we considered.

The conceptual backscattering curve was derived by taking into account both the real observations of S-1 data and the main backscattering mechanisms reported in the literature. In detail, the first phase is related to the initial moistening of the snowpack. During the moistening the value of LWC is low and therefore the SAR backscattering experiences a significant decrease in its value (Shi and Dozier, 1995, Ulaby et al., 2015, Nagler and Rott, 2000, Magagi et al. 2003). During the moistening, the wetting front may be visible only during the afternoon and not in the morning since the snowpack is still subjected to the diurnal cycles of melting and refreezing. As soon as the wetting front has penetrated the superficial insulating layer of the snowpack, the wet snow becomes visible also in the SAR early morning acquisitions with a significant decrease of the backscattering. This condition can be used to identify the start of the snowpack ripening phase. During the ripening phase, which is influenced by the weather and the snowpack conditions, $\sigma^0$ varies according to the snow conditions but with an overall decreasing trend due to the increase of LWC (Shi and Dozier, 1995, Ulaby et al., 2015). We observed that the minimum of $\sigma^0$ is reached in correspondence of the finishing of the ripening phase and the beginning of the run-off phase for all the ten time series observed (see section 5). The run-off phase is instead characterized by a monotonic increase of the backscattering till all the snow is melted. To our knowledge, this characteristic behavior has never been observed in the literature before. Our interpretation is as follows: when the considered snowpack reaches its saturation condition in terms of LWC and snow structure, the backscattering recorded in C-band reaches its minimum value. This snowpack condition seems to correspond with the isothermal condition i.e., the end of the ripening phase. After the saturation point is reached, the monotonic increase of $\sigma^0$ could be

explained by one or the combination of the following factors: i) an increase of the superficial roughness (Shi and Dozier, 1995, Magagi et al. 2003); ii) a change in the snow structure i.e., increase of the density and increase of grain size (Shi and Dozier, 1995, Ulaby et al., 2015) and; iii) at the end of the melting, the presence of patchy snow creates a situation of mixed contribution inside the resolution cell of the SAR and therefore a further increase of the total backscattering is recorded.

From the generalized behavior derived by considering both the observations and the EM background, we derived a set of rules to be applied to the time series of backscattering (that we report in pseudocode in Figure 1b). In order to understand the effectiveness of the proposed rules, they were applied to 1-dimensional cases, made up of the 5 different test sites and the 2-dimensional case of the Zugspitze catchment. In detail, we compare the time onsets derived by the proposed set of rules and the same onsets derived using the algorithm reported in pseudocode in figure 1a from independent measurements of LWC and SWE. This comparison is now discussed in deeper detail as suggested from the Reviewer 2.

The same set of rules has been applied to the 2-dimensional case of the Zugspitze catchment. In this case, a selection of time series of backscattering was randomly selected from the pixels in the Zugspize catchment that were never been used before and reported in Figure 6c of the paper. As one can notice, the characteristic behavior described in section 4.2 is well recognizable.

Finally, it is worth mentioning that from a recent work by Veyssière et al., 2019 (that was not cited in the current version of the manuscript) it is possible to appreciate the classical "U-shape" described in our article derived from an independent dataset ,where S-1 observations were available together with SWE and LWE (simulated). Qualitatively, our proposed rules for the identification of the snow melt phases can be applied also in this independent dataset.
* * *
**Algorithm 1:** Identification of the melting phases

**Input:** Liquid Water Content $LWC$ and Snow Water Equivalent $SWE$ observations for a given day $d$, $d \in \{1, 2, ..., D\}$ with $D$ total number of days with $SWE > 0$, $SWE_{max}$

**Output:** Onset moistening $T_M$, onset ripening $T_R$, onset runoff $T_{RO}$

```
while d ≤ D do
    if LWC_max,d > 0 kg/m² then
        # Snowpack is wet
        # Check moistening phase
        if (LWC_max,d > 1 kg/m²) and (LWC_min,d = 0 kg/m²) for at least 2 days then
            T_M = d
            # Do not check this condition anymore
            continue
        end
        # Check ripening phase
        if (LWC_max,d > 5 kg/m²) and (LWC_min,d > 0 kg/m²) then
            T_R = d
            # Do not check this condition anymore
            continue
        end
        # Check runoff phase
        if (SWE_d == SWE_max) then
            T_RO = d
            # Do not check this condition anymore
            continue
        end
    end
    else
        | # Snowpack is dry
    end
    d ++
end
```

(a)

```
Algorithm 2: Identification of the melting phases
  Input: Multitemporal backscattering observations for different tracks, $\sigma_{morining}$ and $\sigma_{afternoon}$, for
         a given day $d$, $d \in \{1,..,d,..,D\}$ with $D$ total number of observations
  Output: Onset moistening $T_M$, onset ripening $T_R$, onset runoff $T_{RO}$
  while $d \leq D$ do
      if $\sigma_{afternoon,d} - \sigma_{dry} \geq -2\ dB$ then
          # Snowpack is wet
          # Check moistening phase
          if $(\sigma_{morning,d} - \sigma_{dry} < -2\ dB)$ then
              $T_M = d$
              # Do not check this condition anymore
              continue
          end
          # Check ripening phase
          if $(\sigma_{morning,d} - \sigma_{dry} \geq -2\ dB)$ then
              $T_R = d$
              # Do not check this condition anymore
              continue
          end
          # Check runoff phase
          if $(\sigma_d == \sigma_{min})$ then
              $T_{RO} = d$
              # Do not check this condition anymore
              continue
          end
      end
      else
          | # Snowpack is dry
      end
      d ++
  end
```

(b)

**Figure 1** Algorithms used for the identification of the melting phases from: (a) LWC and SWE; and (b) S-1 time series of backscattering.

We agree that a proper EM modeling would strengthen Figure 5, which is now based on real observations and on the literature background. This has been investigated during the work that lead to the present form of the manuscript. Nonetheless, after identifying some crucial limitations of both i) the current RT models in modeling the snowmelt process (especially the runoff conditions as reported later in the review); and ii) the lack of ground truth information on some important snow parameters in the considered test sites (i.e., snow superficial roughness and internal structure during the three melting phases), we favored to present our analysis using the observations of real data and the rich background literature. This in our opinion allowed a quick dissemination of the results obtained in the paper, which has been proved to be reproducible.

However, we would like to show and discuss the SMRT (Picard et al., 2018) results to highlight the limitations of the model to explain the real S-1 behavior of Figure 4 (and therefore of Figure 5), especially during the melting. This is the main motivation why we would like not to include such analysis in the current version of the paper. However, we aim to include the radiative transferring modelling in a future paper. Below we present our preliminary results using the SMRT formulation, which, we believe, confirm the assumption we did in the present paper.

In detail, we consider a simplified one-layer snowpack, derived by averaging the proprieties of the snowpack simulated for Malga Fadner in the hydrological year 2017-2018 by SNOWPACK. Similar conclusions can be repeated for any of the five test sites considered in the paper and for any of the hydrological years. The improved Born approximation (IBA) with sticky hard spheres microstructure, was used together to the discrete ordinate and eigenvalue radiative transfer (DORT) solver in order to simulate

the backscattering coefficient at 5.405 GHz at 34 degrees incidence angle. This SMRT formulation was demonstrated to produce equivalent results of DMRT-based models (Picard et al., 2018).

Following the empirical approach elaborated in Brucker et al., 2010 and Picard et al., 2014, we used non-sticky spheres (i.e., infinite stickiness parameter) and scaled the radius computed from SSA by an empirical factor phi (called "grain size scaling factor"). This was obtained by fitting model results to real Sentinel-1 measurements during the accumulation period. We parametrized the substrate as a reflector providing constant backscattering of -12 dB at VV and -20 at VH (according to what observed in average in dry/frozen conditions).

Figure 2a shows the obtained results. As one can notice SMRT is accurately modeling the backscattering during the accumulation period. But, as soon as the snowpack is getting wet, large differences are visible from the modeled and measured backscattering. The differences are less pronounced in VV than in VH. Interestingly, as in the real S-1 data, an increase in the backscattering is visible after the maximum of SWE is reached. This is mainly due to the grain coarsening during the snowmelt metamorphism. This can be verifying by simulating the same time series with a constant grain size (see Figure 2b). Therefore, it is not clear if there is only a problem of scale (e.g., proper parametrization of the grain size) or some contributions to the backscattering are not considered in model. In particular among the several reasons that can be credited to this behavior we identified:

1. During the snowmelt process, the melt forms tend to group together generating clustered grains with large size;
2. Possible contributions to the total backscattering are not taken into account. In particular, as identified in the manuscript, the contribution by the increasing superficial roughness during the snowmelt should be considered in the model.
3. The value of LWC modeled by SNOWPACK can be overestimated in the considered simulations.

In the following we address in detail the first two points. Whereas for the last point, not having a real ground truth, we can only say that the values seem to be in a reasonable range w.r.t. other alpine snowpacks (Koch et al., 2019, Koch et al., 2014, Heilig et al., 2015, Techel et al., 2011).

As mitigation of point 1, we tried to optimize phi so that to reach the values observed by S-1 during the melting. Even though increasing phi has the effect of increasing the backscattering, it was not possible to converge to a suitable phi. In fact, we reached a point for which the integral of the phase matrix was larger than the scattering coefficient (i.e., the grain size is too big compared to the wavelength before obtaining from the simulation values comparable to the observed values). This suggests that the grain size may be not the only contributor to the total backscattering, and other variables e.g., the superficial roughness, may play a not negligible role.

For point 2 we investigate the possibility to implement the superficial roughness in SMRT. This can be done thanks to the modularity of the code. As discussed in the paper, at the best of our knowledge, only few works have been presented that model the wet snow with active sensors at C-band i.e., Shi and Dozier, 1995; Longepe et al., 2009 and Magagi et al. 2003, Veyssière et al., 2019. Beside the recent work of Veyssière et al., 2019, they are not taking into account the snow microstructure distribution. Even though, in Shi and Dozier, 1995 a deep study of the backscattering mechanisms was conducted with their model, which indicate a positive correlation between largely wet snowpack and the superficial roughness (similar considerations can be found in Magagi et al. 2003), Kendra, Sarabandi and Ulaby, 1998, on the basis of experimental analysis, expressed some doubts on the realism of such model. Therefore, this research topic requires a dedicated effort and validation campaigns that are out of the scope of this paper and it will be

left as future work. This also because continuous measurements of snow roughness are unavailable at the moment.

In summary, for answer the Reviewer question, by using state-of-the-art simulation (i.e., SMRT) it has been possible to partially confirm what already known from the literature, but it did not add a full understanding of the backscattering mechanism, especially in the runoff phase, which in our opinion requires further research. In detail the simulations confirmed:

1. For low amounts of free liquid water in the snowpack, the high dielectric losses increase the absorption coefficient and reduce the recorded backscattering (Shi and Dozier, 1995, Ulaby et al., 2015, Nagler and Rott, 2000).
2. The increasing in grain size has a positive correlation with the volumetric backscattering (Shi and Dozier, 1995, Ulaby et al., 2015), which can be relevant during the melting.
3. The contribution from the ground in general dominates the total backscattering in dry conditions (Rott and Matzler, 1987, Shi and Dozier, 1993) but it is hidden when the snowpack is in wet conditions (Ulaby et al., 2015, Ulaby and et al., 1984).

To incorporate this comment in the paper, section 4.2 has been re-entitled as follows

"Illustrative temporal evolution of backscatter"

Section 5 has been renamed:

"Application of the proposed approach in 1-D and 2-D cases"

and an exhaustive discussion on the limitations of the state-of-the-art model and the lack of validation data during the melting period (e.g., time series of snow roughness) have been added to the paper.

[Figure]

(a)

[Figure]

(b)

**Figure 2** SMRT simulated backscattering coefficient compared with S-1 acquisitions. (a) the grain size has been derived from SNOWPACK simulation; and (b) the grain size has been considered constant to 0.5 mm during all the time interval of the simulation.

Additional references not reported in the manuscript for answering this question.

Brucker, L., Picard, G., Arnaud, L., Barnola, J., Schneebeli, M.,Brunjail, H., Lefebvre, E., and Fily, M.: Modeling time series of microwave brightness temperature at Dome C, Antarctica, usingvertically resolved snow temperature and microstructure mea-surements, J. Glaciol., 57, 171–182, 2011a.

Heilig, A., Mitterer, C., Schmid, L., Wever, N., Schweizer, J., Marshall, H.-P., and Eisen, O. ( 2015), Seasonal and diurnal cycles of liquid water in snow—Measurements and modeling, J. Geophys. Res. Earth Surf., 120, 2139– 2154, doi:10.1002/2015JF003593.

Picard, G., Royer, A., Arnaud, L., and Fily, M.: Influence ofmeter-scale wind-formed features on the variability of the microwave brightness temperature around Dome C in Antarc-tica, The Cryosphere, 8, 1105–1119, https://doi.org/10.5194/tc-8-1105-2014, 2014.

Picard, G., Sandells, M., and Löwe, H.: SMRT: an active–passive microwave radiative transfer model for snow with multiple microstructure and scattering formulations (v1.0), Geosci. Model Dev., 11, 2763–2788, https://doi.org/10.5194/gmd-11-2763-2018, 2018.

Veyssière, G.; Karbou, F.; Morin, S.; Lafaysse, M.; Vionnet, V. Evaluation of Sub-Kilometric Numerical Simulations of C-Band Radar Backscatter over the French Alps against Sentinel-1 Observations. Remote Sens. 2019, 11, 8.

Ulaby, F. T., Stiles, W. H. and Abdelrazik, M., Snowcover Influence on Backscattering from Terrain, in IEEE Transactions on Geoscience and Remote Sensing, vol. GE-22, no. 2, pp. 126-133, March 1984. doi: 10.1109/TGRS.1984.350604

*Specific comments:*

*L.5. Remove "be" in "to be obtained".*

*L.51. Change "The establishing"*

*L.76. change "has demonstrated" to "was shown"*

*L.94-95. This nominal resolution is only true for the high res IW mode. It can be removed in this section of the text since it is better described in the data section.*

*L.98. Remote "the" in "the monitoring"*

*L.106. change "polarimetric" to "polarization"*

*L.144. correct "properties"*

*L.171. remove "round the grains", metamophism does not always round the grains, more complex shapes can be created. Simply remove this part.*

*L.187. change to "October 1, 2016"L.280. correct "removal"*

*L.411-412. not clear to me what you mean by depolarization here. To me depolarized signal implies that the V transmit is switched to H thus increasing VH and decreasing VV. An ice layer alone would not depolarize your signal, it would affect the scattering by adding a reflective layer in your snowpack.*

We thank the Reviewer for pointing out these editorial comments that we will correct in the revised version of the manuscript accordingly.

---

## Author Comment (AC2) · 20 Dec 2019

**Answer to the Anonymous Referee #2 – Manuscript tc-2019-175**

*This is a well-written and clearly organized paper which utilizes the rich new time series of C-band SAR data from Sentinel-1 to explore the extent to which remote sensing can provide snow melt state information for Alpine snowpacks. Detailed analysis of simulations from the physical model SNOWPACK at sites with comprehensive snow state and meteorological measurements allows extension of the approach to broader regions. This provides a realistic approach with respect to operational implementation. I have a number of comments which will hopefully constructively improve the final manuscript.*

We thank the anonymous Reviewer for his/her positive comments.

*1. Line 143: to what extent are C-band measurements influenced by snow grainsize/shape?*

We are thankful to the Reviewer to point out this important issue. As discussed in Section 4.2, the time series of backscattering experiences a monotonic increase after reaching the minimum in correspondence of the maximum of the SWE. We provide three hypotheses for this behavior (Line 420). In particular, in the second hypothesis we suppose the variation of the snow microstructure due to the snowmelt metamorphism i.e., coarsening of the snow grains may contribute to the increase of backscattering. As reported in Section 2.2 of the manuscript (and summarized in Table 1) a positive correlation between the volume scattering and the grain size was found in the literature as a results of experimental observations or the application of EM models (Ulaby, F. et al. 2015, Shi J. and Dozier, J. 1995).

Interestingly, the sensitivity of C-band measurements to the snow grain size and shape can be investigated using the SMRT model (Picard et al., 2018) (indicated by both the Reviewers). This exercise is also useful to understand the validity of the SMRT model in the context of the presented work. In detail, the improved Born approximation (IBA), with sticky hard spheres microstructure, was used together to the discrete ordinate and eigenvalue radiative transfer (DORT) solver in order to simulate the backscattering coefficient at 5.405 GHz at 34 degrees incidence angle. This SMRT formulation was demonstrated to produce equivalent results of DMRT-based models (Picard et al., 2018).

Two typical snowpacks are considered in the simulation: i) a 1.5 m height dry snowpack i.e., liquid water content equal to 0%, density of 250 kg/m3, temperature of -8 °C and; ii) a 0.34 m height snowpack in runoff conditions i.e., liquid water content equal to 4%, density of 450 kg/m3, temperature of 0 °C.

Following the empirical approach elaborated in Brucker et al., 2010 and Picard et al., 2014, we used non-sticky spheres (i.e., infinite stickiness parameter) and scaled the radius computed from SSA by an empirical factor phi (called "grain size scaling factor"). This was obtained by fitting model results to real Sentinel-1 measurements acquired at Malga Fadner at 34 degrees incidence angle. In both the cases we parametrized the substrate as a reflector providing constant backscattering of -12 dB at VV and -20 at VH (according to what observed in average in dry/frozen conditions). Even though there is an implicit relation between grain size and shape and density, in this experiment we keep fixed the density and we vary the radius from 0.1 to 2 mm, corresponding to very fine to coarse in the International classification of snow (Fierz et al., 2009), in order to isolate the main backscattering mechanism due to the grain coarsening.

Fig.1 shows the results of the exercise. As expected, SMRT models a positive correlation between the snow grain size and the backscattering in both dry and wet conditions in accordance with the literature information reported in Section 2.2 of the manuscript. Interestingly, in runoff conditions the simulation is underestimating the real values acquired by Sentinel-1 of several *dB*. As mitigation, we tried to optimize phi for wet snow conditions but it was not possible to converge to a suitable phi since the integral of the

phase matrix was larger than the scattering coefficient i.e., the grain size is too big compared to the wavelength.

[Figure]

[Figure]

(a)                                          (b)

**Figure 1**. Backscattering coefficient simulated by SMRT IBA at 5.405 GHz, 34° incidence angle, phi=4.48, with a constant backscattering of -12 dB at VV and -20 at VH as substrate for: (a) a 1.5 m height dry snowpack i.e., liquid water content equal to 0%, density of 250 kg/m3, temperature of -8 °C and; (b) a 0.34 m height snowpack in runoff conditions i.e., liquid water content equal to 4%, density of 450 kg/m3, temperature of 0 °C. The green and red triangle are the Sentinel-1 VV and VH backscattering measurements, respectively.

If we take the LWC provided by SNOWPACK as correct, this underestimation of the total backscattering may suggest SMRT is not taking into account all the possible contributions to the total backscattering. In particular, as identified in the manuscript (first hypothesis explained in Section 4.2), one of the possible reasons for the observed backscattering increase is the superficial roughness of the snow during the melting (Fassnacht et al., 2009). In fact, at the end of the melting season, especially when it rains over the snowpack, the surface of the snow becomes very rough. An example is illustrated in Figure 2.

[Figure]

**Figure 2**. Example of snowpack at the end of the melting season, during peak runoff phase. The surface is very rough (rms ~ 10 cm). Picture taken by G. Bertoldi and C. Brida in the Venosta Valley in the Italian Alps on 24 June 2019.

When the LWC increases, the absorption coefficient increases, the penetration depth decreases, and the total backscattering is influenced more and more by the superficial characteristics of the snow. In SMRT,

and at the best of our knowledge, in any other models that take into account the EM interactions with the microstructure of the snow, the superficial roughness is generally not modeled. Indeed, in dry conditions, which are the most studied in the literature, the snow surface can be considered smooth (at C-band). As matter of fact, the modeling of the roughness can be performed coupling SMRT with dedicated models such as IEM and its evolutions (Hsieh et al., 1997), nonetheless the testing and the validation of such adaptations requires dedicated effort that is out of the scope of this paper.

It is finally worth saying that, the proper understanding of the backscattering mechanisms during snowmelt is of interest of the authors of this manuscript, both in terms of field observations and EM modelling. This is supported by a recent acceptance of the paper entitled "Identification of multi-temporal snow melting patterns with microwave radars" by Marco Pasian, Pedro Fidel Espín-López, Valentina Premier, Claudia Notarnicola and Carlo Marin to the convened section "Analysis, Design and Use of Microwave Techniques, Models, Systems, and Antennas for Snowpack and Avalanches Monitoring" of the EuCAP conference (i.e., European Conference on Antennas and Propagation). This paper presents the results obtained by a first-order model based on plane-wave on snow by taking into account the parameters that describe the snowpack. The objective is to better understand the driving backscattering mechanisms at the different state of the evolution of a snowpack (especially in terms of snow roughness).

Additional references not reported in the MS for answering this question.

Brucker, L., Picard, G., Arnaud, L., Barnola, J., Schneebeli, M.,Brunjail, H., Lefebvre, E., and Fily, M.: Modeling time series of microwave brightness temperature at Dome C, Antarctica, usingvertically resolved snow temperature and microstructure mea-surements, J. Glaciol., 57, 171–182, 2011a.

Hsieh, C. Y., Fung, A. K., Nesti, G., Sieber, A. J. and Coppo, P. 1997. A further study of the IEM surface scattering model. IEEE Trans. Geosci. Remote Sensing, 35(No. 4) July: 901–909.

Fierz, C., Armstrong, R.L., Durand, Y., Etchevers, P., Greene, E., McClung, D.M., Nishimura, K., Satyawali, P.K., Sokratov, S.A., The international classification for seasonal snow on the ground IHP-VII technical documents in hydrology, 83, IACS Contribution, 1, UNESCO-IHP, Paris (2009).

Picard, G., Royer, A., Arnaud, L., and Fily, M.: Influence ofmeter-scale wind-formed features on the variability of the microwave brightness temperature around Dome C in Antarc-tica, The Cryosphere, 8, 1105–1119, https://doi.org/10.5194/tc-8-1105-2014, 2014.

Picard, G., Sandells, M., and Löwe, H.: SMRT: an active–passive microwave radiative transfer model for snow with multiple microstructure and scattering formulations (v1.0), Geosci. Model Dev., 11, 2763–2788, https://doi.org/10.5194/gmd-11-2763-2018, 2018.

*2. Line 289: "These phases have been identified from the SWE and LWC data according to section 2.1." Were quantitative threshold values of LWC or change in SWE used to identify the three different phases? If so, these are not described in Section 2.1. Some additional detail on how the three melt phases were classified based on the insitu snow measurements would be helpful.*

We thank the reviewer for rising this observation that allows us to better explain this important aspect. Yes, we used a set of objective and reproducible rules to identify the three different phases that lead to the main melting process. These rules have been derived from the considered test sites during the two hydrological years 2016-17 and 2017-18. Even though, they have not been tested under all the possible snowpack

conditions they can be considered general for the alpine snowpacks. In the following we will explain them in detail.

The moistening phase onset is identified by looking at the liquid water content (LWC) of the snowpack. We empirically established a threshold of 1 kg/m$^2$ that has to be satisfied for at least two consecutive days. In other words, a significant melting (and refreezing) cycle should be observed within two days. Among all the isolated moistening events, in this work we focus only on the moistening preceding a ripening phase. However, this does not mean that the SAR cannot detect isolated peaks of melting, if the acquisitions are performed simultaneously to those events.

Regarding the ripening phase, we impose the rule to observe an increase of LWC exceeding 5 kg/m$^2$ and not decreasing to 0 kg/m$^2$ during the diurnal cycles. If the LWC returns to 0 kg/m$^2$ for a timing of at least 5 days, we assume that the ripening phase is interrupted. Otherwise, we assume that there is an enough penetration of the waterfront into the snowpack to initiate the ripening.

Finally, the runoff phase is identified when SWE starts decreasing from its maximum (after the ripening phase is activated). In the case we have both measured and modelled SWE available, we consider measured SWE as reference. The runoff phase ends when SWE has a value of 0 kg/m$^2$.

The rules are shown in the following pseudocode algorithm, which has been added to the revised version of the manuscript.

We would like to point out that Figure 4 shows the three melting phases by considering only the set of rules explained above without taking into account the backscattering behavior.
* * *
**Algorithm 1:** Identification of the melting phases

**Input:** Liquid Water Content $LWC$ and Snow Water Equivalent $SWE$ observations for a given day $d$, $d \in \{1, 2, ..., D\}$ with $D$ total number of days with $SWE > 0$, $SWE_{max}$

**Output:** Onset moistening $T_M$, onset ripening $T_R$, onset runoff $T_{RO}$

```
while d ≤ D do
    if LWC_max,d > 0 kg/m² then
        # Snowpack is wet
        # Check moistening phase
        if (LWC_max,d > 1 kg/m²) and (LWC_min,d = 0 kg/m²) for at least 2 days then
            T_M = d
            # Do not check this condition anymore
            continue
        end
        # Check ripening phase
        if (LWC_max,d > 5 kg/m²) and (LWC_min,d > 0 kg/m²) then
            T_R = d
            # Do not check this condition anymore
            continue
        end
        # Check runoff phase
        if (SWE_d == SWE_max) then
            T_RO = d
            # Do not check this condition anymore
            continue
        end
    end
    else
        | # Snowpack is dry
    end
    d ++
end
```

*3. Section 4.1 provides a detailed description of the S-1 backscatter time series as they relate to snow observations and SNOWPACK simulations. In general, the text provides sufficient explanations for what is shown in Figure 4. This provides a clear observational basis for the synthesis in Section 4.2. My main concern is then the statement on line 385 that "Figure 5 shows the theoretical temporal evolution of backscatter fora complete hydrological year." While the conceptual framework of this figure is based on the measurements in Figure 4, the 'theoretical' component of this figure would be stronger if it contained actual backscatter simulations using a radiometric model (there are numerous options but SMRT comes to mind as a logical choice). I suggest the schematic approach to Figure 5 be augmented with radiometric simulations.*

We are thankful to the Reviewer to point out this crucial issue, which was also raised by Reviewer 1, allowing us to better clarify it. We agree that a proper EM modeling would strengthen Figure 5. So far, this figure should not be considered as a "theoretical" curve, but more as a "conceptual scheme", based on real observations and on the literature background, to better illustrate our interpretation of the backscattering signal in the sites we considered.

The possibility of a proper EM modelling has been investigated during the work that lead to the present form of the manuscript. Nonetheless, after identifying some crucial limitations of both i) the current RT models in dealing with the snowmelt process (especially the very wet snow and runoff conditions); and ii) the lack of ground truth information on snow superficial roughness and internal structure during the three phases, we favored to present our analysis using real data and a (solid) background literature without EM modeling. This allowed a quicker dissemination of the results obtained in the paper. We plan to leave EM modelling for a future step of our research.

However, we would like to show and discuss the SMRT results to highlight the limitations of the model to explain the real S-1 behavior of Figure 4 (and therefore of Figure 5), especially during the melting. This is the main motivation why we decided not to include such analysis in the current version of the paper. To start exploring this issue, we performed the following numerical experiment. In detail, we consider a simplified one-layer snowpack derived by averaging the proprieties of the snowpack simulated for Malga Fadner in the hydrological year 2017-2018 by SNOWPACK. Similar conclusions can be repeated for any of the five test sites considered in the paper and for any of the hydrological years. As done in the sensitivity exercise in response to point 1 of this review, we used non-sticky spheres (i.e., infinite stickiness parameter) and scaled the radius computed from SSA by an empirical factor phi (called "grain size scaling factor"). This was obtained by fitting model results to real Sentinel-1 measurements during the accumulation period. We parametrized the substrate as a reflector providing constant backscattering of -12 dB at VV and -20 at VH (according to what observed in average in dry/frozen conditions).

Figure 2a shows the obtained results. As one can notice, SMRT is accurately modeling the backscattering during the accumulation period. But, as soon as the snowpack is getting wet, large differences are visible from the modeled and measured backscattering. The differences are less pronounced in VV than in VH. As discussed in point 1 of the review, several reasons can be credited to this behavior. In particular:

1. During the snowmelt process, the melt forms tend to group together generating clustered grains with large size;
2. Possible contributions to the total backscattering are not taken into account. In particular, as identified in the manuscript, the contribution by the increasing superficial roughness during the snowmelt should be considered in the model.
3. The absolute value of LWC modeled by SNOWPACK can be overestimated in the considered simulations.

Interestingly, as in the real S-1 data, an increase in the backscattering is visible after the maximum of SWE is reached. This is mainly due to the grain coarsening during the snowmelt metamorphism. This can be verified by simulating the same time series with a constant grain size (see Figure 2b). Therefore, it is not clear if there is only a problem of scale (e.g., proper parametrization of the grain size) or some contributions to the backscattering are not considered in model. A similar underestimation during the melting was noticed in the recent work by Veyssière et al., 2019 (that was not cited in the current version of the manuscript). In this work, it is also possible to appreciate the classical "U-shape" described in our article derived from other test sites where SWE and LWE were simulated. Qualitatively, our proposed rules for the identification of the snow melt phases seem to have a good applicability also in this independent dataset.

As matter of fact, the implementation of the superficial roughness of SMRT can be done thanks to the modularity of the code. However, a better modeling of the EM mechanisms governing the melting process require additional research. Indeed, although wet snow is of great importance for some topics e.g., hydrology, SMRT and any other models that take into account the EM interactions with the microstructure of the snow (i.e., layering and size and distribution of the ice), have been tested and exploited mainly in dry snow conditions. As discussed in the paper, at the best of our knowledge, only few works have been presented that model the wet snow with active sensors at C-band i.e., Shi and Dozier, 1995; Longepe et al., 2009 and Magagi et al. 2003. In particular, they are not taking into account the snow microstructure distribution. Even though in Shi and Dozier, 1995 a deep study of the backscattering mechanisms was conducted with their model, which indicate a positive correlation between largely wet snowpack and the superficial roughness (similar considerations can be found in Magagi et al. 2003), Kendra, Sarabandi and Ulaby, 1998, on the basis of experimental analysis, expressed some doubts on the realist behavior of such a model. Therefore, this research topic requires dedicated efforts and validation campaigns that are out of the scope of this paper and they will be left as future work.

In summary, for answer the Reviewer question, by using state-of-the-art simulation has been possible to partially confirm what already known from the literature. However, in our opinion, further research is required for a full understanding of the backscattering mechanism, especially in the runoff phase. In detail, the simulations confirmed:

1. For low amounts of free liquid water in the snowpack, the high dielectric losses increase the absorption coefficient and reduce the recorded backscattering (Shi and Dozier, 1995, Ulaby et al., 2015, Nagler and Rott, 2000).
2. The increasing in grain size has a positive correlation with the volumetric backscattering (Shi and Dozier, 1995, Ulaby et al., 2015), which can be relevant during the melting.
3. The contribution from the ground in general dominates the total backscattering in dry conditions (Rott and Matzler, 1987, Shi and Dozier, 1993), but it is hidden when the snowpack is in wet conditions (Ulaby et al., 2015, Ulaby and et al., 1984).

To incorporate this comment in the paper, the sentence on line 385 has been modified as follows

"Figure 5 shows the illustrative temporal evolution of backscatter for a complete hydrological year"

and an exhaustive discussion on the limitations of the state-of-the-art model and the lack of validation data during the melting period (e.g., time series of snow roughness) have been added to the paper.

[Figure]

(a)

[Figure]

(b)

**Figure 3** SMRT simulated backscattering coefficient compared with S-1 acquisitions. (a) the grain size has been derived from SNOWPACK simulation; and (b) the grain size has been considered constant to 0.5 mm during all the time interval of the simulation.

Additional reference not reported in the manuscript for answering the question.

Veyssière, G.; Karbou, F.; Morin, S.; Lafaysse, M.; Vionnet, V. Evaluation of Sub-Kilometric Numerical Simulations of C-Band Radar Backscatter over the French Alps against Sentinel-1 Observations. Remote Sens. 2019, 11, 8.

*4. The Conclusion section is quite brief, and does not include a discussion in a number of relevant areas. A couple of suggestions to expand this section:*

We agree with the Reviewer that the conclusion section can be expanded with a more extensive discussion. This has been added in the revised version of the manuscript. In particular the following points raised by the reviewer were addressed.

*4A. There is recent work which suggests SWE can be retrieved from cross-pol C-band SAR measurements, including in the Alps. I agree with your statement on line 150 that "During the accumulation period, dry snow is almost transparent for C-band..." C-band sensitivity to SWE defies a physical explanation. The time series of Sentinel-1 data in your study provide no evidence of sensitivity to SWE (e.g. Figure 4). Can you pro-vide a comment on this in the Discussion, in the context of the work by Lievens et al?https://doi.org/10.1038/s41467-019-12566-y*

The availability of multi-temporal data acquired regularly over the entire globe and made freely accessible opens new opportunities to monitor dynamic phenomena. In particular, monitor snow depth and water equivalent in a systematic and spatially distributed manner would be crucial for a proactive management of the water resources. The paper from Lievens et al., 2019 proposes an empirical algorithm for snow depth retrieval from Sentinel-1 at 1 km resolution. The Authors suggest the retrieval is possible due the cross-polarized information, although the literature on the topic seems to be contradictory. Interestingly all the backscattering time series showed in the paper exhibit the same characteristic "U-shape" identified and analyzed in our work.

Given the easiness of the algorithm proposed in Lievens et al., 2019, we reproduced the change detection algorithm and applied it to our data. Before discussing the results, it is important highlighting some differences in the domain of application of the algorithm:

1. Our data is a restricted and very specific dataset w.r.t. the global one considered in the original paper;
2. We used Sentinel-1 images at the resolution of 20 m, instead of 1 km;
3. We applied a conventional pre-processing procedure to the data, without the empirical normalization of the backscattering for compensating the local incidence angle proposed by the Lievens et al., 2019. This implies that in order to generate a time series of comparable backscattering we can use only the images acquired from the same orbital track. This decreases the number of total images per time series w.r.t. the original work of Lievens et al., 2019.
4. We did not applied any post-processing to the obtained data.

Figure 4 shows the results for each test site and each track. As one can notice for all the considered dataset, the results obtained are poor. RMSE is include between 1.21 m and 4.2 m, the Pearson coefficient R is always lower than 0.56 and the Nash-Sutcliffe efficiency coefficient (NSE or $R^2$) is always negative. One can observe that from the ratio $\sigma_{VH}/\sigma_{VV}$ there are no a clear evidences that the VH polarization is providing information on the HS or SWE. This does not exclude that any of the components mentioned above (especially the higher sampling time) that were not taken into account in this experiment may contribute to increase the sensitivity of the backscattering to some parameters related to the snow height.

In conclusion, the large amount of SAR data made available with a high repetition interval allows the monitoring of the complex processes related to the snow evolution. These processes may produce some effects on the backscattering that may be exploited to indirectly monitor important snow parameters such as the snow depth. We believe this will be one of the most interesting research topic in the future.

[Figure]

(a) Zugspitze Track 117

(b) Zugspitze Track 168

(c) Alpe Tumolo track 95

(d) Alpe Tumolo track 117

(e) Alpe Tumolo track 168

(f) Clozner Loch track 95

(g) Clozner Loch track 117

(h) Clozner Loch track 168

(i) Malga Fadner Track 44

(j) Malga Fadner Track 95

(k) Malga Fadner Track 117

(l) Malga Fadner Track 168

(m) Weissfluhjoch track 15

(n) Weissfluhjoch track 66

(o) Weissfluhjoch track 117

(p) Weissfluhjoch track 168

**Figure 4** Application of the algorithm proposed by Lievens et al., 2019 to the five datasets used in our paper. Upper plot: measured HS and derived HS in orange and blue, respectively. Lower plot: backscattering ratio $\sigma_{VH}/\sigma_{VV}$.

*4B.1. Are the accuracies produced from this study (expressed as the RMSE's on line 495) sufficient to improve current prediction systems used in the Alps?*

The identification of the melting phases was possible for the five considered test sites with an rmse of 6 days for the moistening phase, 4 days for the ripening and 7 days for the runoff phase.

Current operational snow monitoring and/or prediction systems are of different types. They are based on real-time snow ground observations (e.g., WSL Swiss monitoring system https://www.slf.ch/en/avalanche-bulletin-and-snow-situation/snow-maps.html), snow hydrological models (e.g., Mysnowmap for the European Alps https://www.mysnowmaps.com/), remote sensing observations (e.g., ESA *snow_cci* initiative http://cci.esa.int/snow), or the combination of both (e.g., the US National Operational Hydrologic Remote Sensing Center (NOHRSC) https://www.nohrsc.noaa.gov/). The accuracy of such systems varies, but in general is limited by the poor information on snow precipitation, especially in mountain areas. This could lead in errors in estimating snow melt rates and snow disappearance time of several days, even weeks

(Engel et al, 2017). Therefore, the additional information on the snow state and on the runoff onset provided by our approach could be potentially useful to improve the performances of snow monitoring systems.

It is also important to underline that, in order to predict runoff, further hydrological modelling is needed. While the runoff production below the snowpack starts quickly, being snow permeable to water, then the streamflow production can be delayed of several days, even weeks, depending on catchment size and hydrological behavior (Rinaldo et al, 2011).

Therefore, an information of snow runoff onset, even if with an error of about 6 days, can be a very valuable information for an anticipation of the peak stream runoff phase. This can have very important applications for water storage management (e.g., hydropower, irrigation).

Moreover, timely information on the beginning of the moistening process could be relevant for the prediction of dangerous phenomena such as wet-snow avalanches.

*4B.2. Are operational hydrological modeling systems ready to implement the ingestion of snow state estimates from C-band SAR data or are there any impediments?*

Ingestion of remote sensing information for improving snow modelling and monitoring has been extensively applied in the past e.g., Molotch and Margulis, 2008. So far, the most common variable assimilated is snow cover, since this is the most available information acquired using remote sensing.

In our case, we would need to assimilate either information on presence/absence of snow liquid water content or on the snow depletion curve, which can be computed for the first time from the real beginning of the melting (i.e., runoff onset) from high resolution remote sensing data. From the theoretical point of view, this is feasible. However, in the case the assimilation is done on the presence of liquid water content, only snow models, which explicitly simulate free liquid content can be used. These are for example physically based, energy-based snow models such as GEOtop, Amundsen, Crocus or SNOWPACK/ALPINE3D. On the other side, we are confident that the assimilation of the depletion curve calculated from runoff onset derived with the proposed approach would improve the results of spatially distributed simulations.

*4B.3. Since the technique relies on the timing of the backscatter minima to identify the change from ripening to runoff, what is the latency in which a Sentinel-1 derived runoff retrieval could be used given the repeat interval of 6 days (for example, do you need to wait 6 times x number of days to ensure the minima was reached?)?*

We agree that, currently, our approach cannot be used in real time, but we need to wait (at least) for the next acquisition(s) after the minimum. However, we are confident that with a larger database (especially in terms of hydrological seasons available) it would be possible design a more sophisticated and precise algorithm for the identification (or prediction) of the minimum in the backscattering. Moreover, as told before, waiting 6 days could be still enough to anticipate the streamflow runoff peak in large catchments.

Additional references not reported in the manuscript for answering this question.

Molotch, N.P., Margulis, S.A., 2008. Estimating the distribution of snow water equivalent using remotely sensed snow cover data and a spatially distributed snowmelt model: A multi-resolution, multi-sensor comparison. Adv. Water Resour. 31, 1503–1514.

Rinaldo, A., Beven, K. J., Bertuzzo, E., Nicotina, L., Davies, J., Fiori, A., Russo, D., and Botter, G., Catchment travel time distributions and water flow in soils, Water Resour. Res., 47, W07537, doi:10.1029/2011WR010478.

*Editorial: Line 30 – The meaning of the sentence starting with "Precise predictions of the timing..." is not clear. I suggest removing it - the following sentence is more impactful anyway.*

*Line 36: I don't think 'favored by' is the right word choice...'driven by'?*

*Line 46: sentence starting "An increase of LWC..." – this is a very long sentence with many commas. Split into two sentences for readability.*

*Line 49: change to "Continuous measurements of SWE..."*

*Line 502 – if this is in reference to the western United States network of sites, it should be noted as 'SnoTel'.*

*Looks like the panels of Figure 4 are separated by Table 1 and Table 2?*

*Figure 6b requires a legend to indicate the values of the colour scale.*

We thank the Reviewer for pointing out these editorial comments that we correct in the revised version of the manuscript accordingly.

---

## Author Response (AR2)

**Answer to the Anonymous Referee #1 –Manuscript tc-2019-175**

*I really appreciated the thorough work that was done in respect to the first round of reviews. The clarifications provided by the authors answer the questions I had and the limitations of this work are explicitly given in the results/discussion section. This work shows and explains the physical concept of the melting phases of snow in alpine regions and its effects on C-band SAR remote sensing observations.*

We thank the anonymous reviewer for the constructive criticisms that helped to improve the overall scientific quality of the manuscript.

The minor edits suggested by the reviewer have been corrected in the revised version of the manuscripts.

[revised manuscript text omitted]

(a) Zugspitze, season 2016/2017

| | S-1 | Reference | Difference [days] |
|---|---|---|---|
| Moistening | - | 04/04/2018 | - |
| Ripening | 05/04/2018 | 08/04/2018 | -3 |
| Runoff | 18/04/2018 | 18/04/2018 | 0 |

(b) Zugspitze, season 2017/2018

| | S-1 | Reference | Difference [days] |
|---|---|---|---|
| Moistening | 19/03/2017 | 14/03/2017 | +5 |
| Ripening | 23/03/2017 | 20/03/2017 | +3 |
| Runoff 1 | 24/03/2017 | 08/04/2017 | -14 |
| Runoff 1 | 01/05/2017 | 13/05/2017 | -13 |

(c) Alpe del Tumulo, season 2016/2017

| | S-1 | Reference | Difference [days] |
|---|---|---|---|
| Moistening | 07/04/2018 | 02/04/2018 | +5 |
| Ripening | 11/04/2018 | 07/04/2018 | +4 |
| Runoff | 14/04/2018 | 20/04/2018 | -6 |

(d) Alpe del Tumulo, season 2017/2018

| | S-1 | Reference | Difference [days] |
|---|---|---|---|
| Moistening 1 | 23/02/2017 | 14/02/2017 | +9 |
| Moistening 2 | - | 29/04/2017 | - |
| Ripening 1 | 12/03/2017 | 16/03/2017 | -4 |
| Ripening 2 | 28/04/2017 | 05/05/2017 | -7 |
| Runoff 1 | 22/03/2017 | 25/03/2017 | -3 |
| Runoff 2 | 08/05/2017 | 13/05/2017 | -5 |

(e) Clozner Loch, season 2016/2017

| | S-1 | Reference | Difference [days] |
|---|---|---|---|
| Moistening | - | 25/03/2018 | - |
| Ripening | - | 06/04/2018 | - |
| Runoff | 12/04/2018 | 18/04/2018 | -6 |

(f) Clozner Loch, season 2017/2018

| | S-1 | Reference | Difference [days] |
|---|---|---|---|
| Moistening | 19/03/2017 | 14/03/2017 | +5 |
| Ripening | 23/03/2017 | 20/03/2017 | +3 |
| Runoff 1 | 10/04/2017 | 30/03/2017 | +11 |
| Runoff 2 | 07/05/2017 | 09/05/2017 | -2 |

(g) Malga Fadner, season 2016/2017

| | S-1 | Reference | Difference [days] |
|---|---|---|---|
| Moistening | 07/04/2018 | 05/04/2018 | +2 |
| Ripening | 11/04/2018 | 07/04/2018 | +4 |
| Runoff | 21/04/2018 | 19/04/2018 | +2 |

(h) Malga Fadner, season 2017/2018

| | S-1 | Reference | Difference [days] |
|---|---|---|---|
| Moistening | 25/03/2017 | 19/03/2017 | +6 |
| Ripening | 04/04/2017 | 09/04/2017 | -5 |
| Runoff | 14/05/2017 | 16/05/2017 | -2 |

(i) Weissfluhjoch, season 2016/2017

| | S-1 | Reference | Difference [days] |
|---|---|---|---|
| Moistening | 06/04/2018 | 02/04/2018 | +4 |
| Ripening | 10/04/2018 | 17/04/2018 | -7 |
| Runoff | 08/05/2018 | 19/04/2018 | +19 |

(j) Weissfluhjoch, season 2017/2018

**Table 5.** Onset times for the melt phases identified in the five test sites using the LWC and SWE (reference) and Sentinel-1 with the method proposed in the previous section.